# CAPGen: An Environment-Adaptive Generator of Adversarial Patches

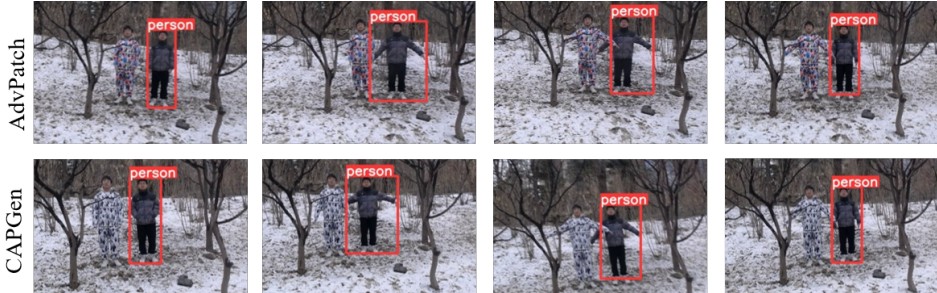

Figure 1: Detection results of different Adversarial coat against Yolov5s. CAPGen-based coat (ours) can better blend with the environment compared to AdvPatch Thys et al. (2019)-based coat.

## Abstract

Adversarial patches, often used to provide physical stealth protection for critical assets and assess perception algorithm robustness, usually neglect the need for visual harmony with the background environment, making them easily noticeable. Moreover, existing methods primarily concentrate on improving attack performance, disregarding the intricate dynamics of adversarial patch elements. In this work, we introduce the **C**amouflaged **A**dversarial **P**attern **Gen**erator (CAPGen), a novel approach that leverages specific base colors from the surrounding environment to produce patches that seamlessly blend with their background for superior visual stealthiness while maintaining robust adversarial performance. We delve into the influence of both patterns (i.e., color-agnostic texture information) and colors on the effectiveness of attacks facilitated by patches, discovering that patterns exert a more pronounced effect on performance than colors. Based on these findings, we propose a rapid generation strategy for adversarial patches. This involves updating the colors of high-performance adversarial patches to align with those of the new environment, ensuring visual stealthiness without compromising adversarial impact. This paper is the first to comprehensively examine the roles played by patterns and colors in the context of adversarial patches.

## 1 Introduction

Adversarial attacks pose a significant challenge in deep learning, aiming to deceive pre-trained models by subtly altering input data like images (Dong et al., 2019), videos (Wei et al., 2022), and text (Ye et al., 2022). These attacks are broadly categorized into digital and physical. Unlike digital attacks that manipulate data within the digital domain, physical attacks interact with the physical environment, presenting unique practical implications and challenges. This distinction underlines the importance of physical attacks, as they have direct consequences for the deployment of deep-learning models in real-world applications, ranging from autonomous driving (Qian et al., 2020) to facial recognition payment systems (Vakhshiteh et al., 2021), thereby elevating the security risks associated with these technologies (Kurakin et al., 2016; Athalye et al., 2018b).

Physical attacks must account for variables such as lighting conditions, angles of view, and physical distances, which can affect the visibility and effectiveness of adversarial inputs. For instance, adversarial patches—a prevalent method in physical attacks—must be designed to maintain their

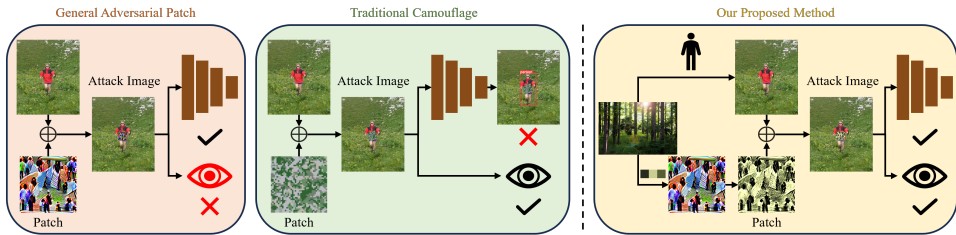

Figure 2: Comparison between adversarial patch (Left), traditional camouflage (Mid) and our proposed method (Right). Adversarial patch can fool AI detector, but is not natural to human. Traditional camouflage(Yang et al., 2020) can fool human observer, but could be detected by AI detectors. Our proposed method can fool human observer and AI detector simultaneously.

deceptive capabilities despite these environmental conditions. However, creating such patches involves overcoming obstacles like ensuring perturbation stealthiness to avoid detection by the human eye, achieving deformation resilience to maintain effectiveness under different physical conditions, and generating adversarial textures that can be quickly and practically applied in diverse real-world scenarios (Brown et al., 2017; Chen et al., 2018).

Recent researches in physical adversarial attacks have extensively explored adversarial patches, demonstrating their potential to compromise machine learning models in real-world scenarios. Notable advancements include AdvPatch (Thys et al., 2019), AdvCloak (Wu et al., 2020), and T-SEA (Huang et al., 2023b), each targeting unique adversarial manipulation aspects. Despite their innovative attack mechanisms, these methods often overlook the importance of patch concealment, making them easily detectable to human observers and thus reducing their practical applicability. Efforts to enhance stealthiness, such as those in (Huang et al., 2020) and (Hu et al., 2021b), have introduced naturalistic adversarial textures, aiming for a dual deception of machine vision systems and human detection. Besides using regular textures for deception, more recent methods consider the background environment by incorporating camouflage or 3D neural rendering. To address visual distortions in existing methods, CamoPatch (Williams & Li, 2024) devises an adversarial patch using semi-transparent RGB-valued circles. Although the designed adversarial texture exhibits good concealability in some specific scenes, its generation process is isolated from the background environment. This causes a noticeable mismatch between the adversarial texture and the background environment in new scenes, diminishing its confusing effect.

As shown in the left of Fig. 2, adversarial patch methods usually do not sufficiently consider the need for adversarial inputs to blend seamlessly into their surroundings or to be quickly and efficiently updated in response to changing environments. This oversight results in adversarial strategies that, while potentially effective in controlled laboratory settings, fall short in complex real-world scenarios, where adaptability is paramount (Duan et al., 2021). In addition to application considerations, there is a lack of in-depth exploration of adversarial patch components, which also limits understanding of physical attacks. Examining the characteristics of adversarial noise in complex and dynamic tasks contributes to the development of more general and robust adversarial patches.

In response to these limitations, we propose a novel adversarial patch generation strategy that prioritizes environmental harmony and rapid adaptability, as shown in the right of Fig. 2. By addressing the critical gaps left by previous methods, our work aims to enhance the robustness and security of machine learning models against physical adversarial threats, contributing valuable insights to the ongoing discourse on machine learning safety and reliability. After formulating the problem by considering both the attack performances and the environmental harmony to human observers, we further propose a method of **C**amouflage **A**dversarial **P**attern **G**enerator (CAPGen) that can generate the adversarial patches effectively.

To improve the visual invisibility of adversarial patches, the specific base colors are extracted from the practical environment, and a patch generation method is designed based on those colors, so as to alleviate the problem of poor visual stealthiness caused by the inconsistency between the patch colors and the environment colors. In more detail, we calculate the probability of each pixel in the adversarial patch for each base color by optimizing a color probability matrix. We ensure that each pixel value corresponds to only one base color by regularizing the color probability matrix. This process allows us to generate an adversarial patch with controllable colors.

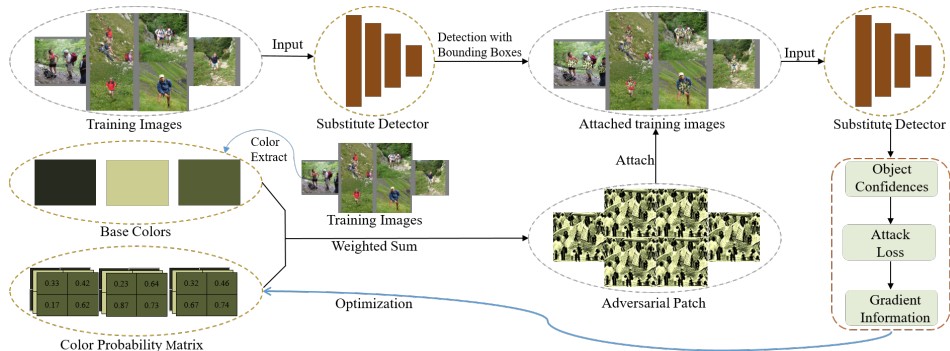

Figure 3: The pipeline of the proposed CAPGen. During the training stage, we optimize a color probability matrix to determine the probability of each pixel corresponding to each base color, which can generate adversarial patches that are consistent with the environment.

We further explore adversarial patch components on attack performance. We assert that adversarial noise influences the predictions of deep models primarily due to the relative magnitude relationship between different pixel values and small perturbations in pixel values. Therefore, the adversarial patch is divided into pattern parts (i.e., color-agnostic texture information) and color parts, and test the model's sensitivity to each part separately. Through comprehensive comparative experiments, we find that patterns have a more significant impact on performance than colors. Based on this finding, we propose a fast adversarial patch generation strategy to adapt to the changing background environments by replacing the colors of high-performance adversarial patches with those matching the environment colors, which can maintain their attack performance while ensuring the patches blend seamlessly into their surroundings. As shown in Fig. 1, we make adversarial patches generated by AdvPatch and CAPGen into coats. We test pedestrians in a snowfield wearing both types of adversarial coats. It is clear that in physical experiments, the adversarial coat generated by CAPGen can better blend with the environment compared to AdvPatch.

The overall algorithmic pipeline is depicted in Fig. 3, our contributions can be summarized as follows:

- We propose a practical algorithm CAPGen that can generate adversarial patches with specified base colors extracted from the environment and has better practicality and invisibility than the mainstream adversarial patch algorithms.

- We start CAPGen by optimizing a color probability matrix, which computes the probability for each pixel in the adversarial patch across base colors. To ensure that each pixel value belongs to just one base color, we apply regularization to the color probability matrix. This process addresses the visually conspicuous issue of the generated patch, making it more harmonious with the environment.

- We pioneer the study of the impact of adversarial patch components (patterns and colors) on attack performance and find that patterns are more significant than colors universally, thus proposing a fast adversarial patch generation strategy. In the black-box setting, with the substitute detector set to Yolov4, the average $mAP_{50}$ of the generated patch on the INRIA dataset even surpasses the mainstream algorithm by about 1.7%. This exploration of patterns and colors have paved a new way for rapid adversarial attacks.

## 2 RELATED WORK

**Physical adversarial attacks.** As mentioned above, the physical adversarial attack is a technology that adds adversarial perturbations to the objects in the physical world to deceive or mislead the deep models. Typically, a physical attack on traffic signs is designed, which achieves deception of detectors by minimizing an adversarial loss function (Song et al., 2018). Thys *et al.* design adversarial patches by attacking the confidence scores of detectors, aiming to evade pedestrian detection systems (Thys et al., 2019). To deal with the impact of distortions brought by the physical space on the attack performance, Expectation Over Transformation (EOT) operations are widely used in various

physical attack algorithms (Athalye et al., 2018b). To get more natural attack effects on object detection systems, clothing with adversarial patches is created (Wu et al., 2020). To alleviate non-rigid deformation, TPS is proposed on clothing (Xu et al., 2020). By replacing adversarial patches with animal shapes, Huang *et al.* achieve more natural attack effects (Huang et al., 2020). To enhance the naturalness of the adversarial clothing, Hu *et al.* extend the adversarial patch to cover the entire clothing using circular topological cropping techniques (Hu et al., 2022). Recent work (Li et al., 2023) is proposed to assess the naturalness of physical attacks. However, the adversarial texture generation process of these methods is isolated from the background environment. This results in a noticeable disparity between the adversarial texture and the background environment in new scenes, diminishing its confusing effect.

**Camouflage.** Traditional camouflage technologies use the color and geometric features of some patterns like stripes or spots to blur and disrupt the edges between the target object and the background image, countering close-range and low-resolution optical reconnaissance (Xue et al., 2016; Owens et al., 2014; Guo et al., 2023; Kajiura et al., 2021). These methods usually take a substantial amount of time to achieve effective camouflage in specific scenarios. Lately, neural radiation field technologies have been used for camouflage by utilizing pre-captured object meshes and multi-view photos (Guo et al., 2023; Zhang et al., 2023). This allows the generation of camouflage for various locations and angles. Obviously, this method requires specific data support and involves intricate 3D object mesh construction. It is worth noting, as shown in the middle of Fig. 2, that traditional camouflage is common means of protection aimed at deceiving human eyes, but lacking of consideration of intelligent models. Recently, Adversarial camouflages are gaining attention as a new means of physical attack, they attack objects in multi-view and multi-scene by rendering the adversarial texture in a 3D simulation environment (Suryanto et al., 2022; Duan et al., 2022; Wang et al., 2022; 2021). They usually use sophisticated networks and some prepared multi-view images to synthesize hard-to-discern object textures. Except for time-consuming optimization, the upfront object mesh is also a substantial workload, making the application relatively challenging.

The above-mentioned methods face different practical challenges, such as poor visual stealthiness, long periods for generating adversarial texture, and complex preparation tasks like 3D modeling or specific simulation environments. This paper prioritizes practicality, focusing mainly on the physical attack mode of adversarial patches. It proposes a more practical and visually concealed means by exploring the generation process of adversarial patches, considering both patterns and colors.

## 3 METHOD

The proposed method is presented in this section. First, we give a problem formulation in Sec. 3.1. Then, we discuss the proposed CAPGen in Sec. 3.2. The fast adversarial patch generation strategy is covered in Sec. 3.3.

### 3.1 PROBLEM FORMULATION

In the domain of adversarial attacks against object detection systems, our investigation is centered on the design of adversarial patches. These are input modifications designed to be placed within a physical scene to compromise the accuracy of object detection models in classifying or detecting objects accurately. This challenge is of particular significance under varying real-world conditions, such as changes in lighting, observation angles, and distances between objects. The formulation involves several core components:

- **Object Detection Model** ($M$): A model parameterized by weights $\theta$, taking an input image $I$ and outputting bounding boxes $B$ and class labels $C$ for detected objects.
- **Target Object** ($O$): The object that the adversary intends to hide from detection or has misclassified by $M$.
- **Adversarial Patch** ($P$): A pixel matrix intended for placement near $O$, engineered to induce detection or classification errors in $M$.

We aim to optimize $P$ to maximize the loss function $L$, causing $M$ to fail in its detection or classification tasks:

$$\max_P \ L(M(I + \delta(P, O, \phi)), O, \theta) \tag{1}$$

where $L$ is the loss function measuring the model's error due to the patch; $\delta(P, O, \phi)$ denotes the application of patch $P$ to image $I$ in the context of the object $O$, with $\phi$ including parameters like the patch's placement, orientation, and scale relative to $O$. Optimizing $P$ requires adhering to constraints that ensure:

- **Robustness to Environmental Variations**: The patch must preserve its adversarial nature under diverse real-world scenarios, necessitating data augmentation to simulate various environmental conditions during development.
- **Stealth and Deception**: Beyond evasion of detection by $M$, $P$ should not attract human attention, ideally blending with its surroundings or appearing as a benign object.

To incorporate these considerations, we propose an advanced optimization framework:

$$\max_P \mathbb{E}_{\phi \sim \Phi} \left[ L(M(I + \delta(P, O, \phi; \epsilon)), O, \theta) \right] - \lambda_1 R(P; \phi) - \lambda_2 S(P; \epsilon) \tag{2}$$

where $\mathbb{E}_{\phi \sim \Phi}$ denotes the expected value over a distribution of conditions $\Phi$ (lighting, angles, distances), encapsulating the robustness to variances; $\delta$ represents the application of the patch $P$ onto image $I$ of object $O$, modified by environmental parameters $\phi$ and stealth parameters $\epsilon$. Moreover, $R(P; \phi)$ quantifies the patch's robustness across varying conditions, encouraging minimal performance degradation; $S(P; \epsilon)$ measures the stealth and deceptive qualities of the patch, ensuring it does not attract undue attention from human observers; and $\lambda_1$ and $\lambda_2$ are regularization coefficients that balance the trade-off between robustness, stealth, and adversarial effectiveness.

This comprehensive formulation outlines a nuanced approach for the development of adversarial patches, emphasizing the need for effectiveness in disrupting object detection algorithms while ensuring resilience to environmental changes and avoidance of human detection. Balancing these factors requires careful consideration of both adversarial goals and the operational context.

## 3.2 Camouflaged Adversarial Pattern Generator

As illustrated in Eq.(2), the stealth and deception of the adversarial patch (controlled by $L(\cdot)$ and $S(\cdot)$, respectively) are contradictory. If $S(\cdot)$ is overemphasized, the patch may lose its adversarial effectiveness and fail to fool the model, and vice versa. Previous methods have neglected the environmental consistency of adversarial patches, which is crucial for realistic attacks. They only concentrate on improving the patches' deception, and thus fail to solve this problem. The key to solving this problem is to find a way to implement $S(\cdot)$ that can achieve stealthiness without affecting the optimization of $L(\cdot)$, that is, separate the optimization processes of $L(\cdot)$ and $S(\cdot)$ into two parallel components as much as possible.

Empirical evidence suggests that humans are more susceptible to being deceived by adversarial patches resembling the background in color and texture. This aligns with studies on human perception, indicating that blending into the surroundings enhances the stealthiness of such attacks. Thus, to make the adversarial patch less noticeable, we first identify the most common colors in the background, which we called base colors and denote by $c$. Simultaneously, we safeguard the patch's adversarial effectiveness through the optimization of a color probability matrix. This matrix, determining the probability of each pixel of an adversarial patch aligning with a base color, undergoes gradient-based optimization. Such an approach decouples the optimization processes for stealth and adversarial effectiveness: base colors ensure visual stealth, while the optimized color probability matrix preserves the patch's adversarial nature.

Specifically, to derive the base colors $c$ from a set of images with similar environmental characteristics, we employ K-means clustering (MacQueen et al., 1967) to partition all pixels within these images into distinct categories. By default, we set the number of categories to 3, unless otherwise specified. Subsequently, we extract the center of each category to form the set $c = \{c_1, c_2, c_3\}$. To obtain the probability of every color in $c$ at each pixel of an adversarial, a color probability matrix $m = [m_{ijk}] \in (0, 1)^{W \times H \times 3}$ is defined, where $W$ and $H$ indicate the width and length of an adversarial patch, respectively. $m_{ijk}$ represents the probability that the color at position (i, j) belongs to the k-th base color. Based on the color probability matrix $m$, the color $t_{ij}$ at position (i, j) can be mathematically as:

$$\begin{cases} t_{ij} = \sum_{k=1}^3 c_k \cdot r_k \\ r_k = \text{Softmax}\left( \frac{\log m_{ijk}}{\tau} \right) \end{cases} \tag{3}$$

where $\tau$ is a temperature coefficient, which controls the blending degree between base colors. We set $\tau$ to 0.1 to keep the color of each pixel belongs to one of the color of base colors. By applying Eq. (3) at each pixel, we can get $P$. We then paste $P$ on objects in images to generate adversarial examples, which we feed into the substitute detector to get detection results. The goal is to fool the model by reducing the confidence scores from these results, thus lowering its detection performance. The pipeline is depicted in Fig. 3.

To improve the robustness of adversarial patches to complex transformations in the real world, such as lighting conditions, rotations, and so on. One common method that we use here is the Expectation Over Transformation (EOT) operations (Athalye et al., 2018a), which adjust the adversarial patch over a range of different transformations in various physical conditions. Specifically, as shown in Eq. (2), we sample various transformations from $\Phi$, including contrast, brightness, noise, angles, and blurring the patch.

### 3.3 FAST ADVERSARIAL PATCH GENERATION

As described in the introduction, there is a lack of in-depth exploration of adversarial patch components, limiting the understanding of physical attacks. To fill the gap, we begin by investigating adversarial patch components, aiming to improve the efficiency of generating adversarial patches through comprehensive comparison and analysis. Physical adversarial attacks are commonly used in real-world situations with limited time and resources. The quick generation and deployment of adversarial patches enhance their practicality across various scenarios. Besides, a prolonged attack process raises the risk of exposure and discovery by security systems or vigilant observers. Efficiency minimizes the time frame of the attack activity, especially in dynamic environments, reducing the chance of detection before successfully attacking the object. It is necessary to improve the stealthiness and efficiency under the adversarial nature in a new scenario.

We assert that adversarial noise influences the predictions of deep models primarily due to the relative magnitude relationship between different pixel values and small perturbations in pixel values. The deep network amplifies these changes in both the relative pixel magnitude and pixel value during an attack, leading to prediction errors. To understand the impact of pixel relative magnitude and color changes on attack performance, we decompose the adversarial patch into two components here: adversarial patterns and adversarial colors. The definitions and operations of these two parts are presented in the subsection. Furthermore, we test the influence of each part on attack performance with the same attack objects and victim models, the experimental results are shown in Sec. 4.3 and Sec. 4.4. Through comprehensive comparative experiments, we find that patterns significantly influence attack performance than colors, and this phenomenon is also universal.

Therefore, a fast adversarial patch generation strategy is proposed based on the above findings. Initially, we create an adversarial patch. Subsequently, when confronted with a new background setting, we substitute the existing colors in the adversarial patch, utilizing base colors from the new background environment. This operation can yield concealed and adversarial patches in a new scenario, circumventing the conventional processes of data collection and adversarial texture optimization. Consequently, it can significantly enhance the efficiency of the adversarial attack.

**Decomposition into Pattern Component.** According to Eq. 3, there exists a set of weights to adjust the blending ratios of base colors, thereby generating different colors. When colors are identical in adjacent regions, no distinct features such as contours or corners emerge, presenting a smooth texture. In contrast, diverse colors in adjacent regions create pronounced boundaries, forming texture information (i.e., semantic details). From the above analysis, we can discern that pattern information is not related to the colors of the individual pixels, but to the differences between the colors of the pixels. Consequently, we formally define the pattern component of an adversarial patch as the relative pixel magnitude.

Due to the color-independence nature of the adversarial patch's pattern, we experiment by randomly substituting its base colors with other different base colors to test its impact on detectors. It is worth noting that we only replace the base colors without altering the color probability matrix. We randomly sample the new base colors from the color space $\mathcal{C} = \{nc_k = (R_k, G_k, B_k) \mid k = 1, 2, 3\}$.

By substituting $nc_k$ for $c_k$ in Eq. (3), the pixel's value at position (i, j) can be formulated as:

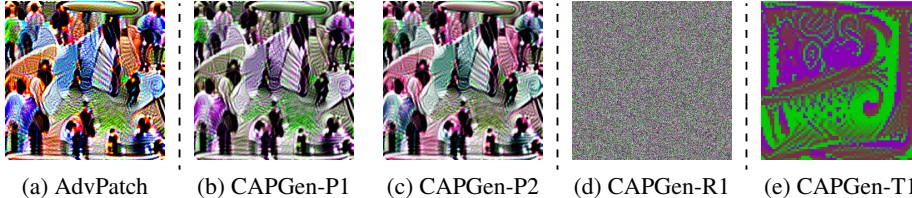

(a) AdvPatch     (b) CAPGen-P1     (c) CAPGen-P2     (d) CAPGen-R1     (e) CAPGen-T1

Figure 4: Adversarial patches obtained by attacking Yolov5s with AdvPatch (4a), its variants after changing base colors (4b, 4c) and its variants after changing color probability matrix (4d, 4e).

$$t_{ij} = \sum_{k=1}^{3} nc_k \cdot r_k \tag{4}$$

As shown in Fig. 4b and Fig. 4c, we can obtain adversarial patches with the same patterns but with different colors. In experiments, we can find that even if we've changed the colors of adversarial patches, we can still achieve powerful attacks.

**Decomposition into Color Component.** We define the color components of adversarial patches as the newly selected base colors $nc_k$ when decoupling the adversarial patch into the pattern components. As demonstrated in Eq. (3), the color probability matrix is still essential for allocating weights among base colors. Thus, to test the impact of color components on detectors, we define two distinct color probability matrices for weight allocation: Color Allocation Based on Random Weights and Gradient Driven Color Allocation.

Color Allocation Based on Random Weights: After determining the base colors, we randomly initialize a color probability matrix $m_r$ to replace the original optimized $m$ in Eq. (3) :

$$r_k = \text{Softmax}\left(\frac{\log(m_r)_{ijk}}{\tau}\right) \tag{5}$$

Gradient Driven Color Allocation: It utilizes gradient optimization to generate adversarial patches while fixing the environmental colors. We use the proposed CAPGen to train the $m_r$, and the trained probability matrix is denoted as $m_t$. The adversarial patches generated by the above two color allocation methods are shown in Fig. 4d and Fig. 4e, respectively.

## 4 EXPERIMENTS

### 4.1 DATASETS AND VICTIM MODELS

We conduct experiments on the INRIA (Dalal & Triggs, 2005) dataset. INRIA is a pedestrian dataset comprising 614 training images and 288 test images. We use Yolov2 (Redmon & Farhadi, 2017), Yolov3 (Redmon & Farhadi, 2018), Yolov4 (Bochkovskiy et al., 2020), Yolov5s, Yolov5m (Ultralytics, 2020), and Faster R-cnn (Ren et al., 2015) as victim models. All these models are pretrained on the COCO dataset(Lin et al., 2014). We use $mAP_{50}$ to measure attack performances, where lower values indicate better attack performance.

### 4.2 BASELINES AND ATTACK SETTINGS

First, we use AdvPatch (Thys et al., 2019) to generate a color-unrestricted adversarial patch. Next, we modify its colors to create color-restricted adversarial patches. We randomly select two base color sets, Bc1 and Bc2, to represent different environments. The RGB values for Bc1 are $[[119, 49, 72], [2, 204, 1], [134, 2, 182]]$, and for Bc2, they are $[[199, 21, 131], [40, 165, 4], [16, 69, 120]]$. We use CAPGen-P1 and CAPGen-P2 to denote adversarial patches created by replacing the colors of the AdvPatch with base colors Bc1 and Bc2. Meanwhile, CAPGen-R1 and CAPGen-R2 are generated using color allocation based on random weights strategy, while CAPGen-T1 and CAPGen-T2 utilize gradient-driven color allocation strategy with the same base colors. To demonstrate the effectiveness of our method, we compare it against two state-of-the-art methods: DAP (Guesmi et al., 2024), NAP (Hu et al., 2021b). Additionally, we also

Table 1: The white-box attack results on the INRIA dataset. The lower the value, the better the attack performance.

| Method | Yolov2 | Yolov3 | Yolov4 | Yolov5s | Yolov5m | Faster R-CNN | Avg. |
|--------|--------|--------|--------|---------|---------|--------------|------|
| Gray | $69.01^{(7.70\downarrow)}$ | $89.20^{(6.70\downarrow)}$ | $83.47^{(8.95\downarrow)}$ | $85.30^{(9.80\downarrow)}$ | $90.20^{(5.10\downarrow)}$ | $73.85^{(7.54\downarrow)}$ | $81.84^{(7.63\downarrow)}$ |
| CAPGen-R1 | $68.12^{(8.59\downarrow)}$ | $90.00^{(5.90\downarrow)}$ | $85.66^{(6.76\downarrow)}$ | $86.30^{(8.80\downarrow)}$ | $89.60^{(5.70\downarrow)}$ | $74.60^{(6.79\downarrow)}$ | $82.38^{(7.09\downarrow)}$ |
| CAPGen-R2 | $68.17^{(8.54\downarrow)}$ | $89.50^{(6.40\downarrow)}$ | $85.29^{(7.13\downarrow)}$ | $85.40^{(9.70\downarrow)}$ | $90.10^{(5.20\downarrow)}$ | $74.60^{(6.79\downarrow)}$ | $82.18^{(7.29\downarrow)}$ |
| CAPGen-T1 | $35.52^{(41.19\downarrow)}$ | $54.10^{(41.80\downarrow)}$ | $63.09^{(29.33\downarrow)}$ | $71.50^{(23.60\downarrow)}$ | $47.00^{(48.30\downarrow)}$ | $17.00^{(64.39\downarrow)}$ | $48.04^{(41.43\downarrow)}$ |
| CAPGen-T2 | $34.71^{(42.00\downarrow)}$ | $56.00^{(39.90\downarrow)}$ | $74.89^{(17.53\downarrow)}$ | $71.70^{(23.40\downarrow)}$ | $34.80^{(60.50\downarrow)}$ | $16.80^{(64.59\downarrow)}$ | $48.15^{(41.32\downarrow)}$ |
| CAPGen-P1 | $14.37^{(62.34\downarrow)}$ | $39.90^{(56.00\downarrow)}$ | $18.15^{(74.27\downarrow)}$ | $32.70^{(62.40\downarrow)}$ | $22.60^{(72.70\downarrow)}$ | $9.80^{(71.59\downarrow)}$ | $\mathbf{22.92}^{(66.55\downarrow)}$ |
| CAPGen-P2 | $16.11^{(60.60\downarrow)}$ | $51.10^{(44.80\downarrow)}$ | $19.63^{(72.79\downarrow)}$ | $36.80^{(58.30\downarrow)}$ | $22.70^{(72.60\downarrow)}$ | $10.11^{(71.28\downarrow)}$ | $26.08^{(63.39\downarrow)}$ |
| DAP | $43.70^{(33.01\downarrow)}$ | $68.07^{(27.83\downarrow)}$ | $20.09^{(72.33\downarrow)}$ | $27.62^{(67.48\downarrow)}$ | $30.06^{(65.24\downarrow)}$ | $63.19^{(18.20\downarrow)}$ | $42.12^{(47.35\downarrow)}$ |
| NAP | $23.8^{(52.91\downarrow)}$ | $57.12^{(38.78\downarrow)}$ | $69.50^{(22.92\downarrow)}$ | $36.94^{(58.16\downarrow)}$ | $64.94^{(30.36\downarrow)}$ | $17.37^{(64.02\downarrow)}$ | $44.95^{(44.53\downarrow)}$ |
| AdvPatch | $10.29^{(66.42\downarrow)}$ | $33.73^{(62.17\downarrow)}$ | $15.53^{(76.89\downarrow)}$ | $31.60^{(63.5\downarrow)}$ | $19.80^{(75.5\downarrow)}$ | $6.50^{(74.89\downarrow)}$ | $\mathbf{19.58}^{(69.89\downarrow)}$ |

introduce a gray patch. For training, we resize all images to $640 * 640$ pixels and set the patch size to 25% of the object's bounding box. The batch size is set to 8, the Adam optimizer with a learning rate of 0.03 is adopted, and the training epoch is set to 200.

## 4.3 WHITE-BOX ATTACK RESULTS

We report the white-box attack results in Tab. 1. The first column lists different attack methods, while the remaining columns present their corresponding results. As shown in Tab. 1, the mean $mAP_{50}$ for CAPGen-P1 is 22.92, significantly lower than the 48.04 for CAPGen-T1, indicating that the model is more vulnerable to patterns. Furthermore, the mean $mAP_{50}$ for CAPGen-P1 is notably lower than that of DAP and NAP by 19.20 and 22.03 points, respectively, highlighting the superior effectiveness of our method in maintaining attack performance. Even AdvPatch is only 3.34 points lower than CAPGen-P1, further illustrating our approach's advantage.

## 4.4 ADVERSARIAL TRANSFERABILITY ANALYSIS

In this section, we report the black-box attack results. Since the white-box attack results for CAPGen-R1 are nearly identical to those for the Gray patch, we chose not to test its black-box performance. The black-box attack results are presented in Tab. 2, where the first column represents the substitute model and the remaining columns represent the target models. As shown in Tab. 2, when using Yolov4 as the substitute model, the mean $mAP_{50}$ for CAPGen-P1 is 37.99, compared to 70.96 for CAPGen-T1. This result suggests that, even in black-box settings, the attack performance of patterns significantly outperforms that of colors. Additionally, the mean $mAP_{50}$ for CAPGen-P1 is nearly the same as that of AdvPatch. For example, when using Yolov4 as the substitute model, the mean $mAP_{50}$ of CAPGen-P1 is 37.99, compared to 38.64 for AdvPatch. This indicates that using a more transferable method as a baseline could further enhance the transferability of our approach. We include additional experimental results based on a more transferable baseline in the supplementary materials.

## 4.5 ABLATION STUDY

**The influence of patch size.** To evaluate how the size of patches affects model performance, we conduct an ablation study on adversarial patch sizes. The size is defined as the percentage of the patch occupying the object area. As shown in the left of Fig. 5, the $mAP_{50}$ for CAPGen-P1 and AdvPatch declines sharply as the size increases. In contrast, the performance of CAPGen-T1 and CAPGen-R1 decreases more gradually. This phenomenon further demonstrates that detectors are more vulnerable to pattern-based adversarial patches.

**The number of base colors.** To evaluate how the number of base colors affects CAPGen's performance, we conduct an ablation study on adversarial patches by gradually increasing the number of base colors. For color selection, we first randomly choose 3 colors, then select an additional 6 colors, resulting in a total of 9 colors. This process continues until we have selected 93 different

Table 2: The black-box attack results on the INRIA dataset. The model of the fist column is held out for the substitute model and the remains are target models.

| Model | Method | Yolov2 | Yolov3 | Yolov4 | Yolov5s | Yolov5m | Faster R-CNN | Avg. |
|---|---|---|---|---|---|---|---|---|
| Yolov2 | CAPGen-T1 | $35.52^{(41.19\downarrow)}$ | $80.40^{(15.50\downarrow)}$ | $81.05^{(11.37\downarrow)}$ | $68.10^{(27.00\downarrow)}$ | $83.90^{(11.40\downarrow)}$ | $65.40^{(15.99\downarrow)}$ | $69.06^{(20.41\downarrow)}$ |
| | CAPGen-P1 | $14.37^{(62.34\downarrow)}$ | $64.70^{(31.20\downarrow)}$ | $60.66^{(31.76\downarrow)}$ | $56.40^{(38.70\downarrow)}$ | $66.30^{(29.00\downarrow)}$ | $34.27^{(47.12\downarrow)}$ | $\mathbf{49.45}^{(40.02\downarrow)}$ |
| | DAP | $43.70^{(33.01\downarrow)}$ | $75.64^{(20.26\downarrow)}$ | $68.82^{(23.60\downarrow)}$ | $50.28^{(44.82\downarrow)}$ | $78.12^{(17.18.00\downarrow)}$ | $51.03^{(30.36\downarrow)}$ | $61.27^{(28.21\downarrow)}$ |
| | NAP | $23.80^{(52.91\downarrow)}$ | $44.41^{(51.49\downarrow)}$ | $46.07^{(46.35\downarrow)}$ | $33.52^{(61.58\downarrow)}$ | $46.88^{(48.42\downarrow)}$ | $28.43^{(52.96\downarrow)}$ | $\mathbf{37.19}^{(52.29\downarrow)}$ |
| | AdvPatch | $10.29^{(66.42\downarrow)}$ | $64.60^{(31.30\downarrow)}$ | $62.64^{(29.78\downarrow)}$ | $57.80^{(37.30\downarrow)}$ | $69.10^{(26.20\downarrow)}$ | $35.57^{(45.82\downarrow)}$ | $50.00^{(36.11\downarrow)}$ |
| Yolov3 | CAPGen-T1 | $61.22^{(15.49\downarrow)}$ | $54.10^{(41.80\downarrow)}$ | $70.61^{(21.81\downarrow)}$ | $70.40^{(24.70\downarrow)}$ | $75.00^{(20.30\downarrow)}$ | $60.70^{(20.69\downarrow)}$ | $65.34^{(24.13\downarrow)}$ |
| | CAPGen-P1 | $42.91^{(33.80\downarrow)}$ | $39.30^{(56.60\downarrow)}$ | $43.72^{(48.70\downarrow)}$ | $50.20^{(44.90\downarrow)}$ | $67.10^{(28.20\downarrow)}$ | $52.90^{(28.49\downarrow)}$ | $49.36^{(40.11\downarrow)}$ |
| | DAP | $45.85^{(30.86\downarrow)}$ | $68.07^{(27.83\downarrow)}$ | $69.07^{(23.35\downarrow)}$ | $56.82^{(38.28\downarrow)}$ | $74.35^{(20.95\downarrow)}$ | $59.03^{(22.36\downarrow)}$ | $62.20^{(27.27\downarrow)}$ |
| | NAP | $40.13^{(36.58\downarrow)}$ | $57.12^{(38.78\downarrow)}$ | $49.05^{(43.37\downarrow)}$ | $30.51^{(64.59\downarrow)}$ | $57.15^{(38.15\downarrow)}$ | $25.48^{(55.91\downarrow)}$ | $\mathbf{43.24}^{(46.23\downarrow)}$ |
| | AdvPatch | $39.56^{(37.15\downarrow)}$ | $33.73^{(62.17\downarrow)}$ | $44.27^{(48.15\downarrow)}$ | $44.1^{(51.00\downarrow)}$ | $63.90^{(31.40\downarrow)}$ | $44.70^{(36.69\downarrow)}$ | $\underline{45.04}^{(42.21\downarrow)}$ |
| Yolov4 | CAPGen-T1 | $53.69^{(23.02\downarrow)}$ | $81.80^{(14.10\downarrow)}$ | $63.09^{(29.33\downarrow)}$ | $73.80^{(21.30\downarrow)}$ | $84.60^{(10.70\downarrow)}$ | $68.78^{(12.61\downarrow)}$ | $70.96^{(18.51\downarrow)}$ |
| | CAPGen-P1 | $25.80^{(50.91\downarrow)}$ | $59.80^{(36.10\downarrow)}$ | $18.15^{(74.27\downarrow)}$ | $39.30^{(55.80\downarrow)}$ | $64.60^{(30.70\downarrow)}$ | $20.27^{(61.12\downarrow)}$ | $\mathbf{37.99}^{(51.48\downarrow)}$ |
| | DAP | $38.47^{(38.24\downarrow)}$ | $58.97^{(36.93\downarrow)}$ | $20.09^{(72.33\downarrow)}$ | $27.33^{(67.77\downarrow)}$ | $58.55^{(36.75\downarrow)}$ | $36.92^{(44.47\downarrow)}$ | $40.06^{(49.42\downarrow)}$ |
| | NAP | $48.27^{(28.44\downarrow)}$ | $59.90^{(43.00\downarrow)}$ | $69.50^{(22.92\downarrow)}$ | $31.15^{(63.95\downarrow)}$ | $57.78^{(37.52\downarrow)}$ | $45.14^{(36.25\downarrow)}$ | $50.79^{(38.68\downarrow)}$ |
| | AdvPatch | $26.51^{(50.20\downarrow)}$ | $62.10^{(33.80\downarrow)}$ | $15.53^{(76.89\downarrow)}$ | $42.90^{(52.2\downarrow)}$ | $62.40^{(32.90\downarrow)}$ | $22.40^{(58.99\downarrow)}$ | $\mathbf{38.64}^{(43.31\downarrow)}$ |
| Yolov5s | CAPGen-T1 | $51.28^{(25.43\downarrow)}$ | $88.00^{(7.90\downarrow)}$ | $85.64^{(6.78\downarrow)}$ | $71.50^{(23.60\downarrow)}$ | $88.50^{(6.80\downarrow)}$ | $72.50^{(8.89\downarrow)}$ | $76.24^{(13.23\downarrow)}$ |
| | CAPGen-P1 | $32.69^{(44.02\downarrow)}$ | $76.70^{(19.20\downarrow)}$ | $86.36^{(6.06\downarrow)}$ | $32.70^{(62.40\downarrow)}$ | $66.30^{(29.00\downarrow)}$ | $36.40^{(44.99\downarrow)}$ | $55.19^{(34.28\downarrow)}$ |
| | DAP | $41.41^{(35.30\downarrow)}$ | $69.74^{(26.16\downarrow)}$ | $66.17^{(26.25\downarrow)}$ | $27.62^{(67.48\downarrow)}$ | $70.91^{(24.39\downarrow)}$ | $43.18^{(38.21\downarrow)}$ | $\underline{53.17}^{(36.30\downarrow)}$ |
| | NAP | $50.40^{(26.31\downarrow)}$ | $63.49^{(32.41\downarrow)}$ | $70.09^{(22.33\downarrow)}$ | $36.94^{(58.16\downarrow)}$ | $65.72^{(29.58\downarrow)}$ | $40.71^{(40.68\downarrow)}$ | $54.56^{(34.91\downarrow)}$ |
| | AdvPatch | $33.02^{(43.69\downarrow)}$ | $75.80^{(20.10\downarrow)}$ | $58.75^{(33.67\downarrow)}$ | $31.60^{(63.50\downarrow)}$ | $62.80^{(32.50\downarrow)}$ | $39.30^{(42.09\downarrow)}$ | $50.21^{(35.02\downarrow)}$ |
| Yolov5m | CAPGen-T1 | $48.82^{(27.89\downarrow)}$ | $61.50^{(34.40\downarrow)}$ | $78.42^{(14.00\downarrow)}$ | $49.00^{(46.10\downarrow)}$ | $47.00^{(48.30\downarrow)}$ | $46.80^{(34.59\downarrow)}$ | $55.26^{(37.21\downarrow)}$ |
| | CAPGen-P1 | $40.35^{(36.36\downarrow)}$ | $47.30^{(48.60\downarrow)}$ | $39.19^{(53.23\downarrow)}$ | $29.90^{(65.20\downarrow)}$ | $22.60^{(72.70\downarrow)}$ | $14.40^{(66.99\downarrow)}$ | $\mathbf{32.29}^{(57.18\downarrow)}$ |
| | DAP | $43.70^{(33.01\downarrow)}$ | $45.75^{(50.15\downarrow)}$ | $47.19^{(45.23\downarrow)}$ | $25.68^{(69.42\downarrow)}$ | $30.06^{(65.24\downarrow)}$ | $41.82^{(39.57\downarrow)}$ | $39.03^{(50.44\downarrow)}$ |
| | NAP | $41.93^{(34.78\downarrow)}$ | $73.96^{(21.94\downarrow)}$ | $62.86^{(29.56\downarrow)}$ | $57.09^{(38.01\downarrow)}$ | $64.94^{(30.36\downarrow)}$ | $50.82^{(30.57\downarrow)}$ | $58.60^{(30.87\downarrow)}$ |
| | AdvPatch | $39.27^{(37.44\downarrow)}$ | $43.50^{(52.40\downarrow)}$ | $38.82^{(53.60\downarrow)}$ | $23.90^{(71.20\downarrow)}$ | $19.80^{(75.50\downarrow)}$ | $12.60^{(68.79\downarrow)}$ | $\mathbf{29.65}^{(53.62\downarrow)}$ |
| Faster R-cnn | CAPGen-T1 | $53.41^{(23.30\downarrow)}$ | $74.20^{(21.70\downarrow)}$ | $81.29^{(11.13\downarrow)}$ | $66.40^{(28.70\downarrow)}$ | $77.00^{(18.30\downarrow)}$ | $17.00^{(64.39\downarrow)}$ | $61.55^{(27.92\downarrow)}$ |
| | CAPGen-P1 | $57.36^{(19.35\downarrow)}$ | $76.60^{(19.30\downarrow)}$ | $71.03^{(21.39\downarrow)}$ | $76.70^{(18.40\downarrow)}$ | $79.50^{(15.80\downarrow)}$ | $9.80^{(71.59\downarrow)}$ | $61.83^{(27.64\downarrow)}$ |
| | DAP | $48.03^{(28.68\downarrow)}$ | $69.57^{(26.33\downarrow)}$ | $79.01^{(13.41\downarrow)}$ | $69.06^{(26.04\downarrow)}$ | $78.52^{(16.78\downarrow)}$ | $63.19^{(18.20\downarrow)}$ | $67.90^{(21.57\downarrow)}$ |
| | NAP | $47.53^{(29.18\downarrow)}$ | $50.20^{(45.70\downarrow)}$ | $66.24^{(26.18\downarrow)}$ | $43.91^{(51.19\downarrow)}$ | $62.45^{(32.85\downarrow)}$ | $17.37^{(64.02\downarrow)}$ | $\mathbf{47.95}^{(41.52\downarrow)}$ |
| | AdvPatch | $51.05^{(25.66\downarrow)}$ | $72.70^{(23.20\downarrow)}$ | $70.61^{(21.81\downarrow)}$ | $74.1^{(21.00\downarrow)}$ | $77.5^{(17.80\downarrow)}$ | $6.50^{(74.89\downarrow)}$ | $\mathbf{58.74}^{(19.91\downarrow)}$ |

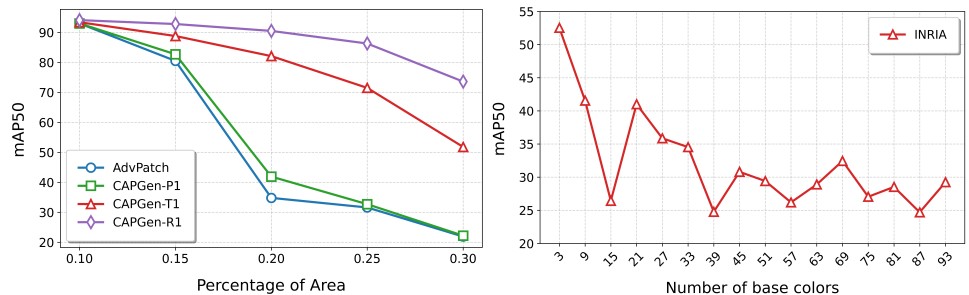

Figure 5: The results of adversarial patches with increasing size (left) and the results with different numbers of base colors (right). All these adversarial patches are generated and tested on Yolov5s.

colors. As shown in the right of Fig. 5, the overall trend in attack performance tends to improve as the number of colors increases.

## 5 CONCLUSION

In this paper, we have proposed a novel algorithm called CAPGen. It can generate patches based on base colors extracted from the environment, thus can fool perception algorithms while being visually concealed in the physical environment. We also explored the effects of patterns and colors on the attack performance of adversarial patches and found that patterns are more important than colors for achieving high attack success rates. Based on this finding, to tackle the long generation period of adversarial patches, we have developed a fast patch generation strategy that can adapt to different environments by changing the colors of existing high-performance patches. The experiments show that our method can generate highly concealed and effective adversarial patches in various scenarios. This work is the first to systematically study the role of patterns and colors in adversarial patch generation, and we hope our findings can open a new path in the field of rapidly generating highly concealed physical attack textures.

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

# APPENDIX

## A    MORE DETECTION RESULTS IN PHYSICAL WORLD

In addition to the physical experiments introduced at the beginning of the paper, as shown in Fig. 1, we also test the proposed method in an additional shrubbery scenario for verifying the stealthiness and effectiveness of our method in different scenarios. The new results are shown in Fig. 6, both the adversarial coats generated by AdvPatch and CAPGen successfully evade detection by Yolov5s. However, the adversarial coat produced by CAPGen can better blend into the environment. We include detection videos in the supplementary materials.

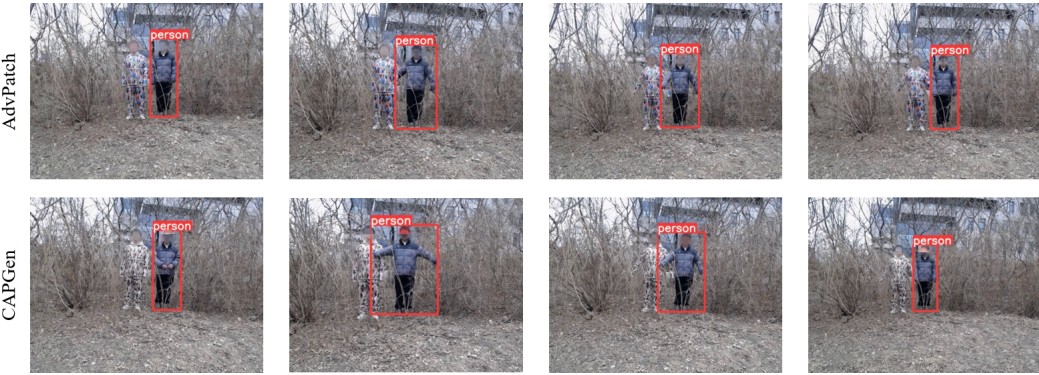

Figure 6: Detection results using AdvPatch and CAPGen in shrubbery. All patches are generated and tested on yolov5s.

## B    WHITE-BOX AND BLACK-BOX ATTACK RESULTS IN FLIR_ADAS DATASET

To verify the effectiveness of our method in different categories, we conduct both white-box and black-box attacks on the FLIR_ADAS dataset. FLIR_ADAS is an autonomous driving dataset containing infrared and visible images, with 10,318 training images and 1,085 test images. In our experiments, we focus on the visible images, resulting in 8,267 training and 859 testing images after excluding those without vehicles. For this dataset, we retrain Yolov2 and Yolov4 to improve performance on clean examples, while the remaining models are pretrained on the COCO dataset. Tab. 3 and Tab. 4 report the results of white-box and black-box attacks, respectively. As shown in Tab. 3, the mean $mAP_{50}$ of CAPGen-P1 is 21.88, while the gray patch is 47.73. This demonstrates that our method can achieve satisfactory attack performance in different categories. Tab. 4 further shows that our method's mean $mAP_{50}$ is similar to that of AdvPatch, indicating that our method can maintain comparable attack performance in different categories.

Table 3: The attacking results on the FLIR_ADAS dataset. The lower the value, the better the attack performance.

| Method | Yolov2 | Yolov3 | Yolov4 | Yolov5s | Yolov5m | Faster R-CNN | Avg. |
|---|---|---|---|---|---|---|---|
| Gray | $30.08^{(3.69\downarrow)}$ | $65.78^{(7.72\downarrow)}$ | $42.69^{(11.84\downarrow)}$ | $60.72^{(6.98\downarrow)}$ | $63.70^{(7.80\downarrow)}$ | $23.42^{(12.49\downarrow)}$ | $47.73^{(8.42\downarrow)}$ |
| CAPGen-R1 | $29.64^{(4.13\downarrow)}$ | $65.34^{(8.16\downarrow)}$ | $42.07^{(12.46\downarrow)}$ | $59.87^{(7.83\downarrow)}$ | $62.84^{(8.66\downarrow)}$ | $24.98^{(10.93\downarrow)}$ | $47.46^{(8.69\downarrow)}$ |
| CAPGen-R2 | $29.55^{(4.22\downarrow)}$ | $65.16^{(8.34\downarrow)}$ | $41.66^{(12.87\downarrow)}$ | $59.60^{(8.10\downarrow)}$ | $62.75^{(8.75\downarrow)}$ | $25.04^{(10.87\downarrow)}$ | $47.29^{(8.86\downarrow)}$ |
| CAPGen-T1 | $12.81^{(20.96\downarrow)}$ | $50.77^{(22.73\downarrow)}$ | $28.38^{(26.15\downarrow)}$ | $39.01^{(28.69\downarrow)}$ | $47.01^{(24.49\downarrow)}$ | $7.42^{(28.49\downarrow)}$ | $30.90^{(25.25\downarrow)}$ |
| CAPGen-T2 | $13.85^{(19.92\downarrow)}$ | $51.26^{(22.24\downarrow)}$ | $30.62^{(23.91\downarrow)}$ | $41.04^{(26.66\downarrow)}$ | $46.09^{(25.41\downarrow)}$ | $7.56^{(28.35\downarrow)}$ | $31.74^{(24.41\downarrow)}$ |
| CAPGen-P1 | $4.31^{(29.46\downarrow)}$ | $42.74^{(30.76\downarrow)}$ | $15.06^{(39.47\downarrow)}$ | $29.30^{(38.4\downarrow)}$ | $38.25^{(33.25\downarrow)}$ | $1.59^{(34.32\downarrow)}$ | $\mathbf{21.88}^{(34.27\downarrow)}$ |
| CAPGen-P2 | $4.31^{(29.46\downarrow)}$ | $44.45^{(29.05\downarrow)}$ | $15.10^{(39.43\downarrow)}$ | $29.38^{(38.32\downarrow)}$ | $39.79^{(31.71\downarrow)}$ | $1.63^{(34.28\downarrow)}$ | $22.44^{(33.71\downarrow)}$ |
| AdvPatch | $3.56^{(30.21\downarrow)}$ | $39.71^{(33.79\downarrow)}$ | $14.89^{(39.64\downarrow)}$ | $24.58^{(43.12\downarrow)}$ | $38.32^{(33.18\downarrow)}$ | $1.00^{(34.91\downarrow)}$ | $\mathbf{20.34}^{(35.81\downarrow)}$ |

Table 4: The black-box attacking results on the FLIR_ADAS dataset. The model of the fist column is held out for the substitute model and the remains are target models.

| Model | Method | Yolov2 | Yolov3 | Yolov4 | Yolov5s | Yolov5m | Faster R-CNN | Avg. |
|---|---|---|---|---|---|---|---|---|
| Yolov2 | CAPGen-T1 | $12.81^{(20.96\downarrow)}$ | $60.48^{(13.02\downarrow)}$ | $31.22^{(23.31\downarrow)}$ | $54.09^{(13.61\downarrow)}$ | $54.75^{(16.75\downarrow)}$ | $14.20^{(21.71\downarrow)}$ | $37.93^{(18.22\downarrow)}$ |
| | CAPGen-P1 | $4.31^{(29.46\downarrow)}$ | $55.99^{(17.51\downarrow)}$ | $21.23^{(33.30\downarrow)}$ | $48.26^{(19.44\downarrow)}$ | $49.76^{(21.74\downarrow)}$ | $6.74^{(29.17\downarrow)}$ | $\mathbf{31.05}^{(25.10\downarrow)}$ |
| | AdvPatch | $3.56^{(30.21\downarrow)}$ | $55.97^{(17.53\downarrow)}$ | $23.19^{(31.34\downarrow)}$ | $47.7^{(20.00\downarrow)}$ | $50.58^{(20.92\downarrow)}$ | $6.8^{(29.11\downarrow)}$ | $\mathbf{31.30}^{(24.85\downarrow)}$ |
| Yolov3 | CAPGen-T1 | $22.48^{(11.29\downarrow)}$ | $50.77^{(22.73\downarrow)}$ | $32.16^{(22.37\downarrow)}$ | $50.84^{(16.86\downarrow)}$ | $50.93^{(20.57\downarrow)}$ | $11.50^{(24.41\downarrow)}$ | $36.45^{(19.70\downarrow)}$ |
| | CAPGen-P1 | $18.69^{(15.08\downarrow)}$ | $42.74^{(30.76\downarrow)}$ | $28.58^{(25.95\downarrow)}$ | $49.20^{(18.50\downarrow)}$ | $46.60^{(24.90\downarrow)}$ | $7.79^{(28.12\downarrow)}$ | $\mathbf{32.27}^{(23.88\downarrow)}$ |
| | AdvPatch | $19.92^{(13.85\downarrow)}$ | $39.71^{(33.79\downarrow)}$ | $30.90^{(23.63\downarrow)}$ | $46.80^{(20.90\downarrow)}$ | $46.29^{(25.21\downarrow)}$ | $7.80^{(28.11\downarrow)}$ | $\mathbf{31.90}^{(24.25\downarrow)}$ |
| Yolov4 | CAPGen-T1 | $18.30^{(15.47\downarrow)}$ | $57.47^{(16.03\downarrow)}$ | $28.38^{(26.15\downarrow)}$ | $52.86^{(14.84\downarrow)}$ | $53.50^{(18.00\downarrow)}$ | $11.37^{(24.54\downarrow)}$ | $36.98^{(19.17\downarrow)}$ |
| | CAPGen-P1 | $11.67^{(22.10\downarrow)}$ | $59.01^{(14.49\downarrow)}$ | $15.06^{(39.47\downarrow)}$ | $48.06^{(19.64\downarrow)}$ | $50.15^{(21.35\downarrow)}$ | $11.03^{(24.88\downarrow)}$ | $\mathbf{32.50}^{(23.65\downarrow)}$ |
| | AdvPatch | $10.54^{(23.23\downarrow)}$ | $58.32^{(15.18\downarrow)}$ | $14.89^{(39.64\downarrow)}$ | $47.21^{(20.49\downarrow)}$ | $50.30^{(21.20\downarrow)}$ | $10.11^{(25.80\downarrow)}$ | $\mathbf{31.90}^{(24.25\downarrow)}$ |
| Yolov5s | CAPGen-T1 | $23.33^{(10.44\downarrow)}$ | $56.11^{(17.39\downarrow)}$ | $33.05^{(21.48\downarrow)}$ | $39.01^{(28.69\downarrow)}$ | $49.45^{(22.05\downarrow)}$ | $8.74^{(27.17\downarrow)}$ | $34.95^{(21.20\downarrow)}$ |
| | CAPGen-P1 | $17.99^{(15.78\downarrow)}$ | $54.58^{(18.92\downarrow)}$ | $25.88^{(28.65\downarrow)}$ | $29.30^{(38.40\downarrow)}$ | $46.66^{(24.84\downarrow)}$ | $2.93^{(32.98\downarrow)}$ | $\mathbf{29.56}^{(26.59\downarrow)}$ |
| | AdvPatch | $18.63^{(15.14\downarrow)}$ | $54.25^{(19.25\downarrow)}$ | $28.45^{(26.08\downarrow)}$ | $24.58^{(43.12\downarrow)}$ | $46.40^{(25.10\downarrow)}$ | $3.90^{(32.01\downarrow)}$ | $\mathbf{29.37}^{(26.78\downarrow)}$ |
| Yolov5m | CAPGen-T1 | $22.31^{(11.46\downarrow)}$ | $55.27^{(18.23\downarrow)}$ | $32.89^{(21.64\downarrow)}$ | $43.29^{(24.41\downarrow)}$ | $47.01^{(24.49\downarrow)}$ | $9.71^{(26.20\downarrow)}$ | $35.08^{(21.07\downarrow)}$ |
| | CAPGen-P1 | $16.52^{(17.25\downarrow)}$ | $51.69^{(21.81\downarrow)}$ | $24.35^{(30.18\downarrow)}$ | $37.24^{(30.46\downarrow)}$ | $38.25^{(33.25\downarrow)}$ | $2.74^{(33.17\downarrow)}$ | $\mathbf{28.47}^{(27.68\downarrow)}$ |
| | AdvPatch | $15.37^{(18.40\downarrow)}$ | $51.83^{(21.67\downarrow)}$ | $25.71^{(28.82\downarrow)}$ | $34.01^{(33.69\downarrow)}$ | $38.32^{(33.18\downarrow)}$ | $3.87^{(32.04\downarrow)}$ | $\mathbf{28.19}^{(27.96\downarrow)}$ |
| Faster R-cnn | CAPGen-T1 | $28.13^{(5.64\downarrow)}$ | $59.58^{(13.92\downarrow)}$ | $39.19^{(15.34\downarrow)}$ | $52.87^{(14.83\downarrow)}$ | $52.35^{(19.15\downarrow)}$ | $7.42^{(28.49\downarrow)}$ | $39.92^{(16.23\downarrow)}$ |
| | CAPGen-P1 | $19.08^{(14.69\downarrow)}$ | $57.56^{(15.94\downarrow)}$ | $27.77^{(26.76\downarrow)}$ | $49.25^{(18.45\downarrow)}$ | $52.99^{(18.51\downarrow)}$ | $1.59^{(34.32\downarrow)}$ | $\mathbf{34.71}^{(21.44\downarrow)}$ |
| | AdvPatch | $19.57^{(14.20\downarrow)}$ | $56.45^{(17.05\downarrow)}$ | $28.57^{(25.96\downarrow)}$ | $49.10^{(18.60\downarrow)}$ | $50.64^{(20.86\downarrow)}$ | $1.00^{(34.91\downarrow)}$ | $\mathbf{34.22}^{(21.93\downarrow)}$ |

## C    MORE DETECTORS ATTACK RESULTS IN DIGITAL WORLD

To demonstrate the consistency of our experimental conclusion, we conduct experiments on two more widely used detectors, namely FCOS (Tian et al., 1904) and RetinaNet (Lin et al., 2017). All these models are pretrained on the COCO dataset(Lin et al., 2014). Following the experimental settings outlined in the main manuscript, we conduct both white-box and black-box attacks. As shown in Tab. 5 and Tab. 6, our proposed method maintains the effectiveness of the original adversarial patches under both white-box and black-box settings and demonstrate the significance of patterns over colors once again.

Table 5: The attacking results on the INRIA and FLIR_ADAS datasets. The lower the value, the better the attack performance.

| Method | Dataset | FCOS | RetinaNet | Avg. | Dataset | FCOS | RetinaNet | Avg. |
|---|---|---|---|---|---|---|---|---|
| Gray | INRIA | $68.93^{(5.29\downarrow)}$ | $68.62^{(6.96\downarrow)}$ | $68.78^{(6.12\downarrow)}$ | FLIR_ADAS | $19.35^{11.99\downarrow}$ | $21.32^{(9.09\downarrow)}$ | $20.34^{(10.54\downarrow)}$ |
| CAPGen-R1 | INRIA | $70.75^{(3.47\downarrow)}$ | $69.89^{(5.69\downarrow)}$ | $70.32^{(4.58\downarrow)}$ | FLIR_ADAS | $21.34^{(10.00\downarrow)}$ | $22.32^{(8.09\downarrow)}$ | $21.83^{9.05\downarrow}$ |
| CAPGen-R2 | INRIA | $68.01^{(6.21\downarrow)}$ | $70.94^{(4.64\downarrow)}$ | $69.48^{(5.42\downarrow)}$ | FLIR_ADAS | $22.24^{(9.10\downarrow)}$ | $22.32^{(8.09\downarrow)}$ | $22.28^{(8.60\downarrow)}$ |
| CAPGen-T1 | INRIA | $6.20^{(68.02\downarrow)}$ | $13.35^{(62.23\downarrow)}$ | $9.78^{(65.12\downarrow)}$ | FLIR_ADAS | $4.73^{(26.61\downarrow)}$ | $6.46^{(23.95\downarrow)}$ | $5.60^{(25.28\downarrow)}$ |
| CAPGen-T2 | INRIA | $6.80^{(67.42\downarrow)}$ | $15.78^{(59.80\downarrow)}$ | $11.29^{(63.61\downarrow)}$ | FLIR_ADAS | $4.85^{(26.49\downarrow)}$ | $8.47^{(21.94\downarrow)}$ | $6.66^{(24.22\downarrow)}$ |
| CAPGen-P1 | INRIA | $3.49^{(70.73\downarrow)}$ | $8.94^{(66.64\downarrow)}$ | $\underline{\mathbf{6.22}}^{(68.68\downarrow)}$ | FLIR_ADAS | $1.84^{(29.50\downarrow)}$ | $2.79^{(27.62\downarrow)}$ | $\underline{\mathbf{2.32}}^{(28.56\downarrow)}$ |
| CAPGen-P2 | INRIA | $3.53^{(70.69\downarrow)}$ | $8.96^{(66.62\downarrow)}$ | $\underline{6.25}^{(68.65\downarrow)}$ | FLIR_ADAS | $1.85^{(29.49\downarrow)}$ | $3.61^{(26.8\downarrow)}$ | $\underline{2.73}^{(28.15\downarrow)}$ |
| AdvPatch | INRIA | $2.26^{(71.96\downarrow)}$ | $3.37^{(72.21\downarrow)}$ | $\mathbf{2.82}^{(72.08\downarrow)}$ | FLIR_ADAS | $1.00^{(30.34\downarrow)}$ | $1.00^{(29.41\downarrow)}$ | $\mathbf{1.00}^{(29.88\downarrow)}$ |

Table 6: The black-box attacking results on the INRIA and FLIR_ADAS datasets. The model of the fist column is held out for the substitute model and the remains are target models.

| Model | Method | Dataset | FCOS | RetinaNet | Avg. | Dataset | FCOS | RetinaNet | Avg. |
|---|---|---|---|---|---|---|---|---|---|
| FCOS | CAPGen-T1 | INRIA | $6.20^{(68.02\downarrow)}$ | $28.16^{(47.42\downarrow)}$ | $17.18^{(57.72\downarrow)}$ | FLIR_ADAS | $4.73^{(26.61\downarrow)}$ | $8.68^{(21.73\downarrow)}$ | $6.71^{(24.17\downarrow)}$ |
| | CAPGen-P1 | INRIA | $3.49^{(70.73\downarrow)}$ | $26.05^{(49.53\downarrow)}$ | $\underline{\mathbf{14.77}}^{(60.13\downarrow)}$ | FLIR_ADAS | $1.84^{(29.50\downarrow)}$ | $6.64^{(23.77\downarrow)}$ | $\underline{\mathbf{4.24}}^{(26.64\downarrow)}$ |
| | AdvPatch | INRIA | $2.26^{(71.96\downarrow)}$ | $21.55^{(54.03\downarrow)}$ | $\mathbf{11.91}^{(62.99\downarrow)}$ | FLIR_ADAS | $1.00^{(30.34\downarrow)}$ | $3.80^{(26.61\downarrow)}$ | $\mathbf{2.40}^{(28.48\downarrow)}$ |
| RetinaNet | CAPGen-T1 | INRIA | $34.25^{(39.97\downarrow)}$ | $13.35^{(62.23\downarrow)}$ | $23.80^{(51.10\downarrow)}$ | FLIR_ADAS | $6.52^{(24.82\downarrow)}$ | $6.46^{(23.95\downarrow)}$ | $6.49^{(24.39\downarrow)}$ |
| | CAPGen-P1 | INRIA | $11.93^{(62.29\downarrow)}$ | $8.94^{(66.64\downarrow)}$ | $\underline{\mathbf{10.44}}^{(64.46\downarrow)}$ | FLIR_ADAS | $4.83^{(26.51\downarrow)}$ | $2.79^{(27.62\downarrow)}$ | $\underline{\mathbf{3.81}}^{(27.07\downarrow)}$ |
| | AdvPatch | INRIA | $12.42^{(61.80\downarrow)}$ | $3.37^{(72.21\downarrow)}$ | $\mathbf{7.90}^{(67.00\downarrow)}$ | FLIR_ADAS | $2.83^{(28.51\downarrow)}$ | $1.00^{(29.41\downarrow)}$ | $\mathbf{1.92}^{(28.96\downarrow)}$ |

## D    IMPLEMENTATION DETAILS OF THE EOT

In this section, we provide a detailed description of EOT. It involves a series of transformations, including adjustments in contrast, brightness, noise levels, rotation angles, and image blurring. Firstly, we use a uniform smoothing kernel $s$ to smooth $m$ and obtain the smoothed matrix $m'$ for reducing the printing error, as shown in below:

$$m' = m * s \tag{6}$$

To reduce the impact of external lighting changes, we implement three specific transformations during the training phase. These include adjustments to contrast ($D \sim \text{Uniform}(d^-, d^+)$), brightness ($B \sim \text{Uniform}(b^-, b^+)$), and noise ($N \sim \text{Uniform}(n^-, n^+)$), where the symbols $*^-$ and $*^+$ denote the lower and upper bounds of the uniform distribution. The outcome of these transformations is a newly formulated color probability matrix $m_p$:

$$m_p = D * m' + B + N \tag{7}$$

Given that attaching an adversarial patch onto an object introduces rotational deformation, we address this challenge by incorporating random rotations of the adversarial patch during training to closely mimic real-world scenarios. Additionally, we employ a scaling matrix designed to alter the size of the adversarial patch, thereby accommodating diverse requirements across different scenarios. The final color probability matrix is expressed as $m_f$:

$$m_f = S * R(\theta) * m_p \tag{8}$$

Here, $R(\theta)$ signifies the rotation matrix responsible for rotating $m_p$ by an angle $\theta$, mathematically defined as:

$$R(\theta) = \begin{bmatrix} cos\theta & sin\theta \\ -sin\theta & cos\theta \end{bmatrix} \tag{9}$$

The scaling matrix $S$, with $S_x$ and $S_y$ representing the scaling ratios along the x-axis and y-axis respectively, is defined as follows:

$$S = \begin{bmatrix} S_x & 0 \\ 0 & S_y \end{bmatrix} \tag{10}$$

In our experiments, We set the contrast transformation to $D \sim \text{Uniform}(0.8, 1.2)$, the brightness transformation to $B \sim \text{Uniform}(0.9, 1.1)$, and the noise transformation to $N \sim \text{Uniform}(0, 0.1)$. The rotation angle $\theta$ ranges from -20 to +20 degrees.

## E    DIFFERENT BASELINES

In the main text, we only use AdvPatch (Thys et al., 2019) as our baseline, to validate the effectiveness of our method, we additionally adopt T-SEA Huang et al. (2023a) as our baseline which has stronger transferability. As shown in Tab. 7, our method can maintain the attack performance of the new baseline both in the white-box and black-box settings while getting better stealthiness.

Table 7: The black-box attacking results on the INRIA dataset.

| Model | Method | Yolov4 | Yolov5s | Faster R-CNN |
|---|---|---|---|---|
| Yolov4 | CAPGen-P1(ours)/T-SEA | 8.14/6.45 | 5.36/4.90 | 16.24/9.84 |
| Yolov5 | CAPGen-P1(ours)/T-SEA | 4.17/4.74 | 2.96/2.38 | 7.70/6.20 |
| Faster R-CNN | CAPGen-P1(ours)/T-SEA | 1.92/1.82 | 2.02/1.94 | 3.03/2.62 |

## F    ADVERSARIAL PATCHES IN BACKGROUND ENVIRONMENTS

To evaluate how well our adversarial patches blend into background environments, we conduct experiments focusing on their visual consistency. First, we adjust the colors of the adversarial patch, generated using AdvPatch, to match the surrounding environments. Then, we directly apply these color-adapted patches to the environments. As shown in Fig. 7, the red circles highlight the locations of the adversarial patches in the images, demonstrating that our patches blend seamlessly with the background.

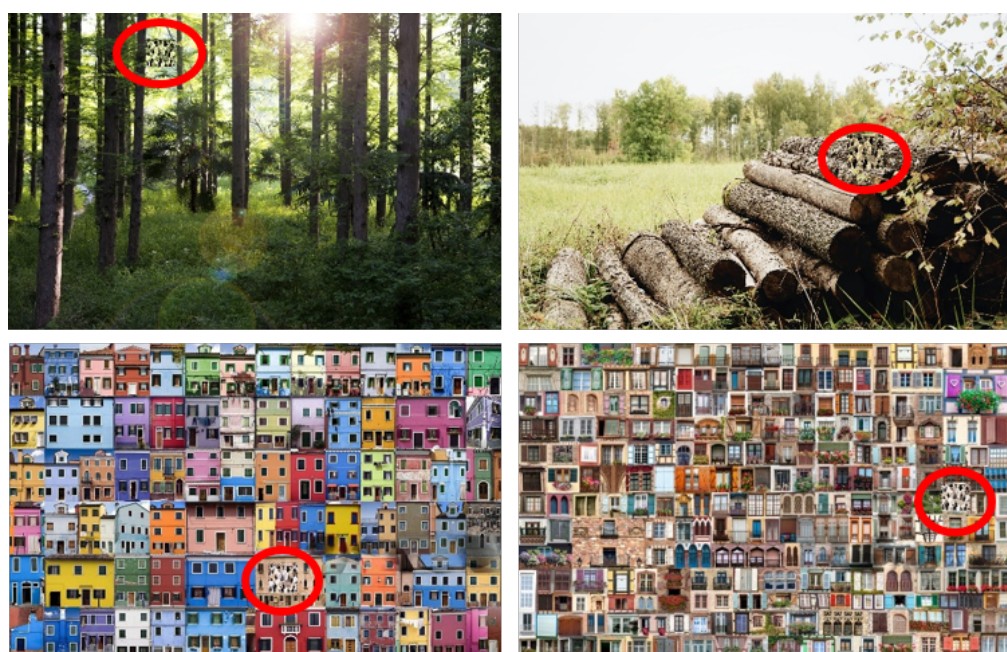

Figure 7: Adversarial patches in background images are marked with red circles.

## G    SUBJECTIVE TEST ABOUT STEALTHINESS OF PATCHES

To quantify the stealthiness of adversarial patches, we subjectively evaluated our patch using a 7-level Likert scale following NAP Hu et al. (2021a). The results are reported in Tab. 8. Compared with the baseline, our method can achieve better stealthiness.

Table 8: Subjective test(1 = not natural at all to 7 = very natural).

| Source | Common texture | AdvPatch | CAPGen-P1(ours) |
|---|---|---|---|
| Score | $5.86 \pm 1.22$ | $1.73 \pm 1.16$ | $4.64 \pm 1.18$ |

## H    VISUALIZATION OF DETECTION RESULTS IN DIGITAL WORLD

To demonstrate the reliability of our experimental findings, we visualize the detection results of different adversarial patches in digital world. As illustrated in Fig. 8 and Fig. 9, the adversarial patch crafted by the AdvPatch shows best effectiveness across both the INRIA and FILA_ADAS datasets. The attack performance of CAPGen-P1 is nearly identical to AdvPatch, while CAPGen-T1 is least effective.

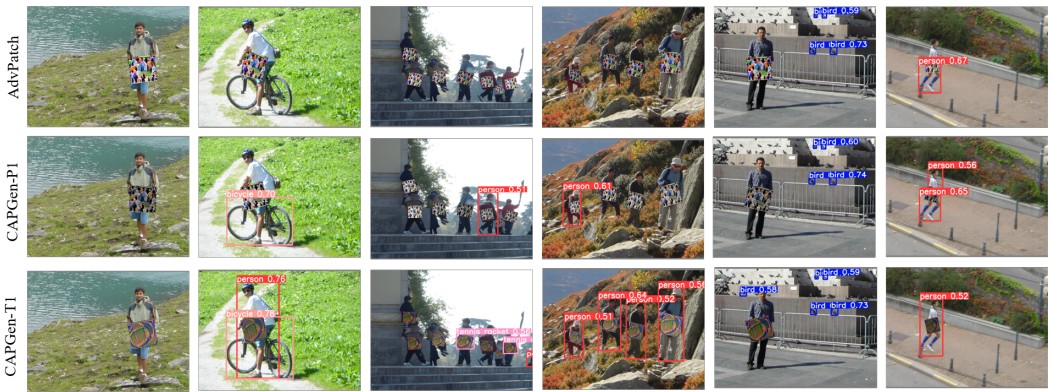

Figure 8: Detection results on INRIA dataset using different adversarial patches. All the adversarial patches are generated and tested on Yolov5s.

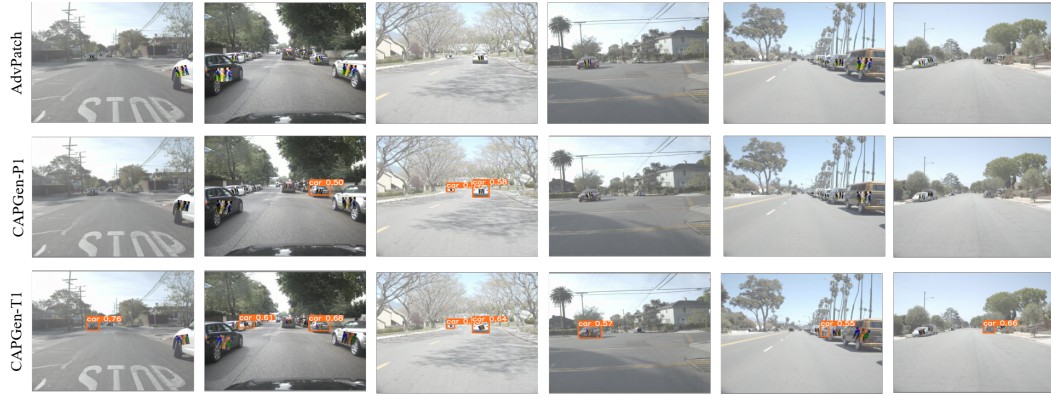

Figure 9: Detection results on FILA_ADAS dataset using different adversarial patches. All the adversarial patches are generated and tested on Yolov5s.

