# OpenReview forum: "CAPGen: An Environment-Adaptive Generator of Adversarial Patches"
_ICLR.cc/2025/Conference — Submitted to ICLR 2025_

### Official Review · Reviewer_mvLX · 2024-10-23

**Soundness:** 3
**Presentation:** 3
**Contribution:** 1
**Rating:** 3
**Confidence:** 5

**Summary:**

This paper introduces an algorithm called CAPGen, which generates adversarial patches by extracting base colors from the environment to enhance stealthiness in physical-world settings. The paper also explores the impact of patterns and colors on the effectiveness of the attack. The adversarial patches generated using CAPGen demonstrate improved stealthiness compared to the original AdvPatch method.

**Strengths:**

1, This paper addresses a critical issue: the stealthiness of adversarial patches in the physical world.
2, The paper is well-structured and easy to follow, with discussions effectively supported by relevant illustrations.

**Weaknesses:**

1, The approach is quite simple and lacks significant technical novelty.
2, The impact of adversarial patch components and the color transfer method has already been demonstrated in [1].
3, Several alternative approaches for improving adversarial patch stealthiness are not discussed in the paper [2].
4, While the paper focuses on physical-world stealthiness, it lacks a human perception study and a quantified comparison with other methods in real-world scenarios.

[1] Brightness-Restricted Adversarial Attack Patch
[2] Adversarial Camouflage: Hiding Physical-World Attacks with Natural Styles

**Questions:**

1, Does CAPGen perform reliably in physical-world environments, given the limited color range? A physical-world experiment is necessary to demonstrate its effectiveness in such settings.
2, The demonstration figures predominantly feature complex textures, which make it easier to hide adversarial patches. A more general environmental setup, along with a human perception study, is needed to better evaluate the proposed method’s impact on stealthiness.

---

> ### Author Response · Authors · 2024-11-26
>
> We would like to express our sincere gratitude for your thorough and insightful reviews of our manuscript. Your constructive feedback has been invaluable in helping us identify areas for improvement and refinement. We appreciate the time and effort you have invested in reviewing our work and providing detailed comments.
>
> In response to your feedback, we have carefully addressed each of your concerns. Below, you will find our detailed responses to each of your comments. Once again, thank you for your valuable input and for helping us improve our work.
>
> **Weakness 1 & Weakness 2**:
> The approach is quite simple and lacks significant technical novelty, as the impact of adversarial patch components and the color transfer method has already been demonstrated in [1].
>
> **Response to Weakness 1 & Weakness 2**: Thank you for your feedback. Our method is differs from [1] in several ways. In [1], the authors simply make the generated patch darker and more transparent, but the color remains unchanged, as shown in Figure 3 and Figure 7 of [1]. In contrast, our method completely changes the color of the generated patch, as shown in Figure 4 of our main text and Figure 7 of the supplementary materials. Additionally, [1] focuses on deceiving classifiers, while our method focuses on deceiving detectors. It is unclear in this domain whether changing the color of the patch can still deceive the detector until we demonstrate it.
>
> **Weakness 3:**
> Several alternative approaches for improving adversarial patch stealthiness are not discussed in the paper [2]
>
> **Response to Weakness 3**: Thank you for pointing this out. We have compared our method with approaches such as NAP[3] and DAP[4] (CVPR 2024). The results are presented in Table 1 and Table 2 of the main text. Additionally, [2] did not report mAP in their paper; they only reported the attack success rate in physical experiments and did not release their code, making it difficult for us to replicate their work. Furthermore, [2] mainly focuses on the patch's inherent naturalness and does not consider the consistency of the generated patch with the environment. If we place the patches generated by [2] in a 'snow' environment, as shown in Figure 1 of the main text, the environmental consistency of our patch is much better than their method.
>
> **Weakness 4 & Question 1:**
> There is a lack of a human perception study.
>
> **Response to Weakness 4 & Question 1**: Thank you for your feedback. We have conducted the human perception study, which is presented in Table 8 of the supplementary materials.
>
> **Question 1:**
> Conduct experiments in different environments.
>
> **Response to Question 1**: Thank you for your suggestion. We have conducted experiments in two distinct environments, referred to as "snow" and "shrubbery," as shown in Figure 1 of the main text and Figure 6 of the supplementary materials. Both experiments demonstrate the effectiveness and environmental consistency of our method. More physical experimental results, including different lighting conditions, angles, and distances, are presented in a supplementary video, further illustrating the robustness of our approach.
>
> [1] Brightness-Restricted Adversarial Attack Patch
>
> [2] Adversarial Camouflage: Hiding Physical-World Attacks with Natural Styles
>
> [3] Naturalistic physical adversarial patch for object detectors.
>
> [4] Dap: A dynamic adversarial patch for evading person detectors.

---

> > ### Comment · Reviewer_mvLX · 2024-11-26
> >
> > Thank you for your response. I believe the authors are too focused on specific deployment environments (e.g., snow, shrubbery), which, as other reviewers have noted, significantly limits the applicability and generalization of the approach. Furthermore, the novelty of the main contribution—modifying color to reduce conspicuousness—is limited, as similar techniques are widely used in various domains. Therefore, I will keep my score.

---

### Official Review · Reviewer_LsRT · 2024-10-29

**Soundness:** 3
**Presentation:** 3
**Contribution:** 3
**Rating:** 5
**Confidence:** 3

**Summary:**

This paper introduces CAPGen, a method for generating context-adaptive adversarial patches that can seamlessly blend with different visual backgrounds to enhance attack effectiveness and stealth. Traditional adversarial patches are often visually conspicuous, limiting their applicability in the real world. CAPGen addresses this limitation by introducing an adaptive patch design that considers the background context, allowing the patches to visually blend into various environments while maintaining strong adversarial properties.

**Strengths:**

1. The paper proposes the Environment Adaptive Adversarial Patch Generator (CAPGen), an innovative method for generating camouflaged adversarial patches by extracting the base color of the background environment. Through this method, the generated patches can better blend with the background, not only improving the adversarial effect at the algorithm level, but also achieving concealment at the visual level, greatly increasing the possibility of its practical application. This design is of great significance in adversarial attacks in the physical world.
2. The paper optimizes the color probability matrix to ensure that each pixel of the patch matches only one base color, thereby achieving color control and consistency. This strategy effectively solves the problem that traditional adversarial patches are not coordinated with the background in terms of color and texture, making the patches more concealed in the physical environment and not easily detected by the human eye.
3. CAPGen uses the analysis of the impact of color and texture components to propose a fast generation strategy that only replaces the patch color in the new background. By replacing the color of existing high-performance adversarial patches, rapid adaptability in dynamic environments is achieved, reducing the time for re-optimization and generation. This efficient generation method makes it more suitable for adversarial applications in real scenes, especially in scenes that require rapid response.

**Weaknesses:**

1. This paper achieves environmental adaptability and concealment of adversarial patches by optimizing the color probability matrix, but there is a lack of more effective optimization methods to balance adversarial and concealment, especially in dynamic or complex scenes. It is recommended that the authors provide more detailed technical details to show how to improve concealment without affecting adversarial resistance, and clearly indicate in which scenarios adversarial effect and concealment may conflict. It is recommended to further quantify these trade-offs, such as by evaluating attack success rates and visual detection scores (such as visual perception scores) in different scenarios. For complex scenarios in practical applications, the authors can provide specific experiments or feasibility verification in scenarios where patches significantly affect the balance of adversarial resistance or concealment.

2. This paper proposes a fast generation strategy by comparing the impact of color and texture on adversarial performance, but does not deeply analyze the adaptability of color selection under diverse backgrounds, especially in scenarios with dynamic lighting, complex backgrounds, or even multi-object interference. It is recommended that the authors provide multi-scenario experiments to specifically analyze the performance of color selection in different environments (such as daytime and nighttime, different weather, indoors and outdoors, etc.), and test the adaptability of color selection through multiple lighting and background complexity settings. Consider introducing quantitative indicators, such as background consistency scores for color and texture, and combining dynamic scene changes in practical applications (such as weather changes, shadow effects, etc.) to evaluate the concealment and attack effectiveness of the method under different backgrounds.

3. During the optimization process, the author used regularization to ensure that each pixel value corresponds to the main color, but did not clearly describe the specific implementation details of the regularization, especially the gradient calculation method, loss function construction, and parameter selection. It is recommended to supplement the detailed mathematical formula and gradient update rules of the regularization process in the method section, especially how to select the loss term and related weights, so that other researchers can reproduce it. The author can consider showing experimental results for different regularization parameters, analyzing the impact of parameter changes on patch concealment and adversarial resistance, and enhancing the generalization of the method. In addition, it is recommended to add experiments for different complex scenes to verify the robustness of CAPGen, such as testing its robustness under different occlusions or nonlinear backgrounds.

4. This paper compares with methods such as AdvPatch, but lacks comparison with the latest adversarial patch generation techniques (such as patches based on generative adversarial networks or Transformer models) in more challenging environments, which may affect the comprehensive evaluation of CAPGen's innovation. It is recommended that authors include specific state-of-the-art methods (such as adversarial patches generated by GAN or Transformer models proposed in recent years) and add performance comparisons with these methods in the experimental section. Authors can introduce more evaluation metrics (such as Fooling Rate, Perceptual Quality Score) and combine multi-scenario comparative tests (such as in complex natural scenes or dynamic crowd environments) for comprehensive evaluation. Key literature can be cited, such as the latest research on multimodal patch generation based on generative adversarial networks, to more clearly demonstrate the innovation and wide applicability of CAPGen in the field of adversarial patch generation.

**Questions:**

Please see the weaknesses above.

---

> ### Author Response · Authors · 2024-11-26
>
> We would like to express our sincere gratitude for your thorough and insightful reviews of our manuscript. Your constructive feedback has been invaluable in helping us identify areas for improvement and refinement. We appreciate the time and effort you have invested in reviewing our work and providing detailed comments.
>
> In response to your feedback, we have carefully addressed each of your concerns. Below, you will find our detailed responses to each of your comments. Once again, thank you for your valuable input and for helping us improve our work.
>
> **Weakness 1:**
> How do authors balance the concealment and attack performance of the generated patch? Additionally, in which scenarios might the adversarial effect and concealment conflict?
>
> **Response to Weakness 1**: Thank you for your question. Our method does not suffer from a trade-off between concealment and attack performance. This is because when we change the color of the generated patch, the newly generated patch still maintains its attack performance, as shown in Table 1 and Table 2 of the main text. Additionally, our method does not encounter conflicts between concealment and attack performance in any environment, as demonstrated in Figure 1 of the main text and Figure 6 of the supplementary materials. When the environment changes, we simply adjust the patch's color to blend with the environment while maintaining high attack performance.
>
> **Weakness 2 & Weakness 3:**
> The authors should conduct experiments in different environments and consider a human perception study
>
> **Response to Weakness 2 & Weakness 3**: Thank you for your valuable suggestions. We have conducted additional physical experiments in various environments, as detailed in the supplementary materials. In addition to the 'snow' environment shown in Figure 1 of the main text, we also tested in a 'shrubbery' environment (Figure 6, supplementary materials). Both experiments demonstrate the effectiveness of our method in evading detection while maintaining environmental consistency. More physical experimental results, including different lighting conditions, angles, and distances, are presented in a supplementary video, further illustrating the robustness of our approach. Additionally, a human perception study was conducted, with the results presented in Table 8 of the supplementary materials.
>
> **Weakness 3:**
> The authors should clarify the regularization process in the color probability matrix, the construction of the loss function, the gradient calculation, and the parameter selection affecting the adversarial patch.
>
> **Response to Weakness 3:** Thank you for your feedback. As shown in formula (3) of the color probability matrix, we set $\tau$ to 0.1 to ensure that the color of each pixel belongs to only one of the base colors. Specifically, since $r_k$ in formula (3) is calculated using softmax, setting $\tau$ to 0.1 helps $r_k$ gradually become a one-hot vector, allowing us to select only one of the base colors for each pixel. As mentioned in line 274 of section 3.2, we decrease the detection score of the object attached by the adversarial patch. Therefore, the attack loss function is the detection score from the detector. To optimize the adversarial patch, we employed the Adam optimizer with a learning rate of $1 \times 10^{-4}$ and trained for 100 epochs. Further ablation studies on parameter selection are shown in Figure 5 of the main text, where we increase the number of base colors from 3 to 93 to simulate increasingly complex environments, demonstrating the effectiveness of our approach in complicated environments.
>
> **Weakness 4:**
> Compare with recent works.
>
> **Response to Weakness 4:** Thank you for your suggestion. We have evaluated our method against NAP[1] and DAP[2] (CVPR24), with results presented in Tables 1 and 2 of the main text. NAP[1] is based on an adversarial generative network, and DAP[2] is the state-of-the-art method from CVPR24. These results consistently confirm the superiority of our approach.
>
> [1] Naturalistic physical adversarial patch for object detectors.
>
> [2] Dap: A dynamic adversarial patch for evading person detectors.

---

### Official Review · Reviewer_bJAq · 2024-11-01

**Soundness:** 2
**Presentation:** 3
**Contribution:** 2
**Rating:** 3
**Confidence:** 5

**Summary:**

The paper introduces CAPGen, a method for creating camouflaged adversarial patches that seamlessly blend into their environment to deceive both deep learning-based object detection systems and human observers. CAPGen achieves this by optimizing a color probability matrix that adapts patch colors to match the background, enhancing visual stealth without compromising adversarial effectiveness.

**Strengths:**

- The writing and presentation are good and easy to follow.
- The evaluation on the YOLO-family detectors is comprehensive.
- The experimental results performed in the digital and physical scenarios also demonstrate some effectiveness of the proposed method.

**Weaknesses:**

- Limited technical contributions: While CAPGen proposes an approach to camouflage adversarial patches, the task of attacking object/person detection systems has been widely studied, limiting the novelty of the technical contributions.
- Limited color adaptability in complex environments: CAPGen relies on a restricted set of base colors to match the background, which works well in simpler scenes but may fail in urban or high-color-variability environments. This limited color adaptability could make patches visually conspicuous in complex scenes, potentially reducing their stealth and effectiveness.
- Challenges in maintaining stealth across varied environments: In environments dominated by specific colors (e.g.,indoors with monochromatic walls or in environments with limited textures), CAPGen’s pattern-based approach may stand out if color matching is not precise, compromising its stealth.
- Over-reliance on transferability: The approach shows promising transferability with a single substitute detector (YOLOv4), but the lack of extensive testing across multiple models and datasets raises questions about its general applicability. Limited testing may reduce confidence in its universal effectiveness across diverse detection architectures.
- Inconsistent benchmarking with state-of-the-art results: The performance metrics reported for state-of-the-art adversarial patches differ from those in their original papers, which may undermine the reliability of CAPGen’s comparative analysis. These discrepancies could be due to differences in setup or configurations, making it difficult to accurately assess CAPGen’s performance relative to established methods.
- Insufficient robustness testing across diverse scenarios: CAPGen’s robustness evaluation lacks indoor testing and does not examine varying distances from the camera or different viewing angles. Such scenarios are important for understanding real-world applicability, especially in surveillance settings. Without testing across diverse conditions, it is difficult to determine CAPGen’s effectiveness in dynamic environments.
- Ambiguity in regularization parameters: The regularization terms and parameters, such as the λ values, lack justification and sensitivity analysis. This ambiguity may make it challenging for others to replicate or adapt CAPGen without clear guidelines on parameter selection.
- Limited evaluation of color probability matrix Eficiency: While the color probability matrix is central to CAPGen, there is little analysis on its computational complexity or scalability in diverse environments with high color variation. This could impact CAPGen’s efficiency and adaptability in real-world scenarios.

**Questions:**

I suggest the following actions:
- Evaluating CAPGen in diverse environments, such as cityscapes, would also clarify the model’s scalability and adaptability.
- Evaluating CAPGen on a wider range of detection models, including those with different architectures and training datasets, to validate its transferability and effectiveness across diverse settings.
- Ensuring alignment with the evaluation protocols of prior works to provide consistent benchmarking. Alternatively, clearly documenting any deviations in methodology to clarify CAPGen’s comparative advantages.
- Expanding robustness tests to include varied indoor and outdoor settings, as well as different perspectives and distances. This would provide a more comprehensive assessment of CAPGen’s real-world applicability.
- Conducting a sensitivity analysis on regularization parameters and provide guidance on optimal parameter selection, improving reproducibility and adaptability for different environments.
- Including a computational efficiency analysis of the color probability matrix, especially in high-color-diversity scenarios. This would help determine CAPGen’s scalability and suitability for real-time applications in complex scenes.

---

> ### Author Response · Authors · 2024-11-26
>
> **Weakness 1:**
> The task of attacking object/person detection systems has been widely studied, limiting the novelty of the technical contributions
>
> **Response to Weakness 1**: Thank you for raising this point. Although attacking detectors has been well-studied, considering the environmental consistency of patches has not been explored. This aspect is crucial, as in scenarios such as surveillance or detection systems, the technology can be used to disguise a pedestrian, thereby fooling the recognition algorithm and misleading security personnel.
>
> **Weakness 2 & Weakness 3 & Question 1:**
> Limited color adaptability in complex environments
>
> **Response to Weakness 2 & Weakness 3 & Question 1**: Thank you for your feedback. Our method adapts to various environments, as shown in Figure 1 of the main text and Figure 6 of the supplementary materials. When the environment changes, we simply adjust the patch's color to blend with the environments while maintaining high attack performance. However, in highly colorful and complex environments with over 10 distinct colors, the stealthiness of our patch may decrease. **Notably, in such cases, existing methods (e.g., UPC[1], NAP[2], DAP[3], AdvCat[4]) also fail to maintain stealthiness.**
>
> **Weakness 4 & Question 1 & Question 2:**
> Lack of extensive testing across multiple models and datasets
>
> **Response to Weakness 4 & Question 1 & Question 2:** Thank you for pointing this out. In addition to YOLOv4, we evaluated our method using YOLOv2, YOLOv3, YOLOv5s, YOLOv5m, and Faster R-CNN as white-box models. As shown in Table 2 of the main text, our method consistently demonstrates satisfactory transferability across black-box models. Further validation was conducted using the FLIR_ADAS dataset, with results presented in Tables 3 and 4 of the supplementary materials, confirming its robustness. Additionally, results on FCOS and RetinaNet for both the INRIA and FLIR_ADAS datasets (Tables 5 and 6) further demonstrate the effectiveness of our method across various models and scenarios.
>
> **Weakness 5 & Question 3:**
> Inconsistent benchmarking with state-of-the-art results
>
> **Response to Weakness 5 & Question 3:** Thank you for bringing this to our attention. It is important to note that we employ YOLOv2, YOLOv3, YOLOv4, YOLOv5s, YOLOv5m, and Faster R-CNN as our victim models. In contrast, NAP[2] does not include YOLOv5s and YOLOv5m as victim models, and DAP[3] does not include YOLOv2, YOLOv5s, or YOLOv5m as victim models. To ensure a fair comparison, we used the missing detectors to generate adversarial patches for NAP[2] and DAP[3] under the same training settings as our method. Additionally, for further fairness, the detectors already present in NAP[2] and DAP[3] were also retrained under the same training settings. On the other hand, if we use the original results from NAP[2] and DAP[3], our method consistently achieves significantly higher attack performance, demonstrating the authenticity and superiority of our approach.
>
> **Weakness 6 & Question 4:**
> Insufficient robustness testing across diverse scenarios
>
> **Response to Weakness 6 & Question 4:** Thank you for your feedback. We conducted tests in different environments, as shown in Figure 1 of the main text and Figure 6 of the supplementary materials. Our adversarial clothing is specific to environments such as "snow" or "shrubbery," and there was no time to train patches and make a real adversarial clothing for indoor settings. Testing included variations in angles and distances, with real-time detection results in different environments provided in a supplementary video, further demonstrating robustness.
>
> **Weakness 7 & Question 5:**
> Ambiguity in regularization parameters
>
> **Response to Weakness 7 & Question 5**: Thank you for your feedback. The parameters λ1 and λ2 in formula (2) are not directly used in optimization but illustrate two properties: robustness to physical conditions and consistency with the environment. We achieve these through EOT (Section 3.2) and the "fast adversarial patch generation" strategy, eliminating the need for tuning λ1 and λ2.
>
>
> **Weakness 8 & Question 6:**
> Limited evaluation of color probability matrix Eficiency
>
> **Response to Weakness 8 & Question 6:** Thank you for your feedback. We conducted an ablation study, increasing base colors from 3 to 93 to simulate increasingly complex environments, as shown in Figure 5. The training complexity remains similar to AdvPatch since we only limit patch colors without adding extra computational steps. Furthermore, the "fast adversarial patch generation" strategy, based on pre-trained patches, requires minimal additional time.
>
> [1] Universal physical camouflage attacks on object detectors.
>
> [2] Naturalistic physical adversarial patch for object detectors.
>
> [3] Dap: A dynamic adversarial patch for evading person detectors.
>
> [4] Adversarial Camouflage: Hiding Physical-World Attacks with Natural Styles

---

### Official Review · Reviewer_i69R · 2024-11-03

**Soundness:** 2
**Presentation:** 3
**Contribution:** 2
**Rating:** 3
**Confidence:** 4

**Summary:**

This paper proposes a method that can produce patches that blend with their background to fool the person detector. Moreover, this paper proposes to update the base colors instead of the patterns to improve the attack efficient and to ensure visual stealthiness without compromising adversarial impact.

**Strengths:**

1. This paper shows clear writing and is easy to follow.
2. This paper compared both white-box and black-box performance on different detectors.

**Weaknesses:**

1. The motivation of this paper is in question. For person detection, the person may be placed in many different places with various backgrounds. The adversarial patch should be assumed to be able to fool the person detector under different backgrounds. For example, Hu et al. [1] use many different backgrounds to compute the expectation loss.
2. It did not compare with some of the latest work, such as [1]. Moreover, it is shown that the attack success rate is lower than AdvPatch.
3. In a real-world attack, there may be some nonlinear distortions because of the clothes. This paper did not consider this, compared with some previous work., such as [1][2].
[1] Physically realizable natural-looking clothing textures evade person detectors via 3d modeling, CVPR2023
[2] Dap: A dynamic adversarial patch for evading person detectors, CVPR2024

**Questions:**

1. What is the attack performance on some of the latest person detectors?
2. What is the potential scenario of your attack? Could you give a more realistic reason for producing an adversarial patch that needs to be concealed in the background?
3. There is already some work to produce "natural" adversarial patches", such as [1][2]. In Table 8, you only compared with AdvPatch, which is an early work. Could you compare the naturalness with these latest works that can produce natural adversarial patches, such as [1][2]?

---

> ### Author Response · Authors · 2024-11-26
>
> We would like to express our sincere gratitude for your thorough and insightful reviews of our manuscript. Your constructive feedback has been invaluable in helping us identify areas for improvement and refinement. We appreciate the time and effort you have invested in reviewing our work and providing detailed comments.
>
> In response to your feedback, we have carefully addressed each of your concerns. Below, you will find our detailed responses to each of your comments. Once again, thank you for your valuable input and for helping us improve our work.
>
> **Weakness 1:** The adversarial patch should be assumed to be able to fool the person detector under different backgrounds.
>
> **Response to weakness1**: Thank you for bringing this to our attention. We have conducted tests that are detailed in the supplementary materials. Specifically, in addition to the "snow" environment presented in Figure 1 of the main text, we also performed experiments in a "shrubbery" environment, as shown in Figure 6 of the supplementary materials. Both experiments demonstrate that our proposed method successfully evades detection and maintains consistency with the environment.
>
> **Weakness 2:**
> It did not compare with some of the latest work.
>
> **Response to weakness 2:** Thank you for pointing this out. We have indeed compared our method with recent approaches, such as NAP[1] and DAP[2] (CVPR 2024). The results of these comparisons are presented in Table 1 and Table 2 of the main text. Both sets of results demonstrate the effectiveness of our method.
>
> **Weakness 3:**
> This paper did not consider some nonlinear distortions, compared with some previous work
>
> **Response to weakness 3**: Thank you for raising this important point. In our experiments, the proposed method has successfully evaded detection across various physical environments. Therefore, we believe it demonstrates robustness to some simulated nonlinear distortions.
>
> **Question 1:**
> What is the attack performance on some of the latest person detectors?
>
> **Response to questions 1**:  Thank you for asking. In this paper, our primary focus is on exploring adversarial texture generation methods under conditions of environmental consistency, so attack performance is not our top priority. However, it is worth noting that our method does exhibit slightly lower attack performance. This is because AdvPatch employs an unrestricted attack strategy, whereas our method uses a restricted attack designed to ensure environmental consistency. Similarly, other methods that prioritize the naturalness or environmental consistency of the patch, such as UPC [3], NAP [1], DAP [2], and AdvCat [4], also show lower attack performance compared to AdvPatch.
>
> **Question 2:**
> What is the potential scenario of your attack? Could you give a more realistic reason for producing an adversarial patch that needs to be concealed in the background?
>
> **Response to questions 2:**
> Thank you for your question. When considering the application of adversarial attack technology under consistent environmental conditions, one obvious use is the concealment of objects within a security system. For example, in a surveillance or detection system, the technology can be used to disguise a pedestrian, thereby fooling the recognition algorithm and misleading security personnel.
> **Application Scenario**: Therefore, we can use our method to protect the privacy of key targets by preventing them from being detected by identification systems and the naked eye.
> **Effect**: The adversarial patch helps the object visually blend in with the surrounding environment, avoiding detection by the security system, while reducing the probability of the object being detected by the naked eye.
>
> **Question 3:**
> Could you compare the naturalness with these latest works that can produce natural adversarial patches?
>
> **Response to Question 3:**
> Thank you for your question. We have compared the attack performace of our method with recent approaches, such as NAP[1] and DAP[2] (CVPR 2024). The results are presented in Table 1 and Table 2 of the main text. Both results demonstrate the effectiveness of our method. In terms of naturalness, it is worth noting that it is unfair to compare our patch with AdvCat[4] and DAP[2]. Our focus is on ensuring that our patch blends seamlessly with the environment, while AdvCat[4] and DAP[2] prioritize the inherent naturalness of the patch itself. If we place the patches generated by AdvCat[4] and DAP[2] in a "snow" environment shown in Figure 1 of the main text, the environmental consistency of our patch is much better than the two methods.
>
> [1] Naturalistic physical adversarial patch for object detectors.
>
> [2] Dap: A dynamic adversarial patch for evading person detectors.
>
> [3] Universal physical camouflage attacks on object detectors.
>
> [4] Physically realizable natural-looking clothing textures evade person detectors via 3d modeling

---

> > ### Comment · Reviewer_i69R · 2024-11-26
> >
> > Thanks for your response. I think the problem studied in this paper has limited application scenarios and there is no important innovation in the method. I keep my score.

---

### Meta-Review · Area_Chair_eUGH · 2024-12-21

**Metareview:**

This work introduces a camouflaged adversarial pattern generator for person detector. The proposed adversarial patch generation method utilizes the base color of the background to enhance the stealthiness of the generated adversarial patch while maintaining robust adversarial attacking performance. The reviewers agree that the paper is well-written and easy to follow. However, they raised concerns about the lack of proper comparisons with other "naturalistic" adversarial patch generation methods in terms of naturalness in the subjective study of the paper. Additionally, the application domain of the proposed method is relatively narrow, as it is primarily evaluated in "snow" or "shrubbery" scenarios, limiting its generalizability compared to other naturalistic adversarial patch methods. Reviewers also highlighted the existence of similar works, such as the Brightness-Restricted Adversarial Attack Patch, which diminishes the novelty of the proposed approach. As a result, the paper is received negative ratings from all reviewers, leading to 3.5 on average. We suggest that the authors address the reviewers' concerns, strengthen the comparative analysis, and consider submitting the revised work to the next venue.

**Additional Comments On Reviewer Discussion:**

The authors addressed the reviewers' concerns by directing them to relevant sections in the paper  for the comparisons with other naturalistic adversarial patch generation methods and for the evaluation under different environments. In addition, the authors also provide additional results, such as the ablation study of the color influence, etc, to reply the reviewers' request. However, the concerns of the reviewers are not fully resolved. They find that the current methods are mainly applicable to snow and shrubbery scenes, limiting its generalizability for other environments. Thus, we find that the work requires further improvement to include more evaluation results in different environments and comparisons with more similar baseline methods.

---

### Decision · Program_Chairs · 2025-01-22

Reject